# Gaps in Monitoring Leave Northern Australian Mammal Fauna with Uncertain Futures

**Noel Preece** [1,2,*] and **James Fitzsimons** [3,4]

1 Tropical Environmental & Sustainability Science (TESS), College of Science & Engineering, James Cook University, Cairns, QLD 4870, Australia
2 Research Institute for the Environment and Livelihoods, Charles Darwin University, Darwin, NT 0909, Australia
3 The Nature Conservancy, Suite 2-01, 60 Leicester Street, Carlton, VIC 3053, Australia; jfitzsimons@tnc.org
4 School of Life and Environmental Sciences, Deakin University, 221 Burwood Highway, Burwood, VIC 3125, Australia
* Correspondence: noel@biome5.com.au

**Abstract:** Northern Australian biomes hold high biodiversity values within largely intact vegetation complexes, yet many species of mammals, and some other taxa, are endangered. Recently, six mammal species were added to the 20 or so already listed in the Australian endangered category. Current predictions suggest that nine species of mammal in northern Australia are in imminent danger of extinction within 20 years. We examine the robustness of the assumptions of status and trends in light of the low levels of monitoring of species and ecosystems across northern Australia, including monitoring the effects of management actions. The causes of the declines include a warming climate, pest species, changed fire regimes, grazing by introduced herbivores, and diseases, and work to help species and ecosystems recover is being conducted across the region. Indigenous custodians who work on the land have the potential and capacity to provide a significant human resource to tackle the challenge of species recovery. By working with non-Indigenous researchers and conservation managers, and with adequate support and incentives, many improvements in species' downward trajectories could be made. We propose a strategy to establish a network of monitoring sites based on a pragmatic approach by prioritizing particular bioregions. The policies that determine research and monitoring investment need to be re-set and new and modified approaches need to be implemented urgently. The funding needs to be returned to levels that are adequate for the task. At present resourcing levels, species are likely to become extinct through an avoidable attrition process.

**Keywords:** mammals; population decline; threatened species; feral species; fire; grazing; disease; conservation; Indigenous land management

## 1. Introduction

An avalanche of research shows that global faunal declines and extinctions have increased over recent decades [1–3] and none of the Convention on Biological Diversity's Aichi biodiversity targets had been met by the end of the decade for biodiversity [4,5]. Governments around the world have largely failed to produce or to implement policies, and to provide the necessary resources to effectively arrest and reverse the biodiversity declines [5,6]. In northern Australia, mammal species continue to decline [7], despite a mostly intact vegetation [8]. While much biodiversity loss can be attributed to direct human impacts, they are not always good predictors [9,10]. While there is no doubt that some species in northern Australia are in decline and heading towards extinction, the broad assessment of extinction rates is based on limited monitoring across the region, so many extrapolations and assumptions have been made. The monitoring of species and ecosystems are necessary for the conservation and management of species and their habitats [11,12]. Monitoring must be statistically robust in order to draw correct conclusions about species

declines [13], to anticipate the thresholds of significant change, and to understand what and how processes and drivers are affecting decline [2].

Most of the 1.9 million square kilometers (26% of the Australian landmass) [14] of the northern Australian savanna region lacks any monitoring of species or ecosystems. This is in part due to a reduction in research and monitoring capacity through the demise of institutions, such as Commonwealth Scientific and Industrial Research Organisation (CSIRO) centers, that are devoted to ecosystem and species research, the limited capacity of research institutions (only two universities with an ecological research focus exist in northern Australia), the reliance on remote sensing and vegetation condition monitoring [15], which do not detect faunal trends, and the dilution of focused research by state environment departments coinciding with dispersed research efforts from Natural Resource Management (NRM) groups and private conservation organizations. The research and monitoring efforts have declined significantly, in part due to a 50% reduction in federal environment funding since 2013 [16,17], equivalent reductions in state and territory environment funds (e.g., 20% cuts in Queensland environment budget in 2012 [18]), and the loss of periodic funding that had supported biodiversity survey and monitoring programs over the past decade or so. Some research on species trends is being conducted under the National Environmental Science Program, but funding is relatively low (149 million AUD in 2021 over six years), considering that the program covers Australia. These losses have resulted in a history of ad hoc, intermittent, and ephemeral survey and monitoring programs that have failed due to the lack of support by governments and scientists, the lack of interest from journals in publishing articles on monitoring, and the difficulties of undertaking research and monitoring [19].

This article reviews and synthesizes the research on and the monitoring of threatened vertebrate species of northern Australia, with an emphasis on small to medium size (35–5500 g) mammals [20]. We first reviewed the literature on the biodiversity of northern Australian savannas, with an emphasis on that occurring between 2010 and 2021, and interviewed many of the ecologists working in the savannas in order to identify additional published articles, grey literature, and active monitoring projects and programs. We each maintain comprehensive bibliographies of research on the fauna in northern Australia from our combined six decades of working in the region and so, with the additional benefit of our close links with research colleagues and practitioners in the region, a strict primary 'literature review' was deemed unnecessary. The purpose of the review is to (1) update and present the range of issues around faunal declines in the region in order to provide context to the gaps in monitoring; (2) articulate the gaps in knowledge in order to better frame the future monitoring of species and ecosystems; and (3) suggest an approach to improve monitoring and research, identifying some of the priority areas and the scale of the resources needed. The study updates and extends previous reviews [21–23]. Although the lack of monitoring for many mammal species across Australia has been documented before [19], we seek to articulate the problem at a regional level in order to identify the locations and the extent of the gaps in biodiversity monitoring sites across north Australia specifically, and to build on suggestions for biodiversity monitoring in relation to prescribed burning outcomes [24] with a view to operationalizing monitoring.

## 2. Mammal Decline in Northern Australia

### 2.1. Background

The extinctions and declines in apparently intact landscapes in Australian arid and semi-arid zones are well documented [7,20,25], but despite more than a century of records and studies, the status of biodiversity across these regions, and of the tropical savannas of northern Australia, is relatively poorly known. Since the 1990s, dramatic declines in mammal fauna richness and abundance have been observed in the northern part of the Northern Territory (NT) and the Kimberley region of Western Australia [7,26–28] but the understanding of the trends across the region is based almost entirely on monitoring for around three decades in just three national parks in the NT [22,23,29–31], plus a number of autecological studies of species [32–35]. The currency of declines has been reinforced recently in Kakadu National Park [31,36]. Declines have also been occurring in the Queensland tropical savannas (half of the savanna region) for decades [37,38], but these have not been well monitored (Box 1) and there are no recently published studies of status and trends in Queensland.

**Box 1.** Queensland tropical savannas as a particular case.

---

The biodiversity studies conducted over the past decade in north Queensland on ten different properties (60 sites) showed alarmingly low trapping rates for small mammals. For example, on Olkola people's country, an area known as the Kimba Range, produced a trapping rate of 0.17 mammal species per site (range 0.08–0.71), and all of the sites that were studied across the Northern Gulf and Cape York regions produced similarly low trap success (N. Preece, unpubl. data). This rate was considered to be 'extremely low' for Nitmiluk in the NT [30] and in Kakadu National Park, mean species richness per site declined in 15 sites from about 2.7 to 0.5 over 13 years [29] and was considered 'alarming'.

The number of 'mammal-empty' sites or plots was also telling, as 84% of the sites on the Kimba Range recorded no mammals (N. Preece, unpubl. data). Other studies of some 202 sites across the Cape York region produced similarly low trap success, with the proportion of sites that recorded no mammals ranging from 0.5% to 47.5% [39]. These figures are of concern as studies of Kakadu mammal fauna, which produced 55% mammal-empty sites, which was considered to be extremely poor and showed a significant downward trend from 13% mammal-empty sites 13 years previously (1996–2009), indicating a rapid and severe decline in the mammal fauna in that location [29].

The abundance of mammals per site in other north Queensland surveys was also extremely poor when compared with other studies in the savanna region [27,29,40]. Of particular concern is that otherwise common species of small mammal, such as the delicate mouse *Pseudomys delicatulus*, and several native rat species, which are found normally in most fauna surveys across the region and are well adapted to sparsely vegetated environments [6,41], were almost completely absent from these surveys.

---

We focus mostly on mammalian fauna as most of the declines have been of mammals, particularly those in the 'critical weight range' (35 g to 5.5 kg), although not exclusively so [22,23,42–44]. Some northern monsoonal savanna bird fauna are threatened e.g., [29,45] but generally do not show the same declining trends as mammals [31,46,47], although some species are of greater concern than others [48]. Several reptile species are showing declines in north Australia, apparently more so than in southern regions [49]. Three are critically endangered (*Austroblepharus barrylyoni*; *Bellatorias obiri*; *Lerista allanae*), two endangered (*Saltuarius eximius*; *Lerista ameles*), one vulnerable (*Orraya occultus*), and one is considered by expert elicitation to be vulnerable (*Lerista storri*) [49], but most reptiles seem to be secure [50].



*2.2. Recent Changes*

Since the review by Ziembicki et al. (2015) [23] a number of changes to the status of mammals have occurred, due to the recognition of some subspecies in the legislation in different Australian jurisdictions and to changes in the knowledge and status of other species. We have updated Table 1 from Ziembicki et al. (2015) [23] to reflect these changes and have added the status of species under state and territory legislation (Table 1). Four rock-wallaby taxa, the Kimberley nabarlek *Petrogale concinna monastria*, the Top End nabarlek *P. c. canescens*, the Cape York rock-wallaby *P. coenensis*, and the West Kimberley rock-wallaby *P. lateralis kimberleyensis* have been newly listed as endangered under Australia's *Environment Protection and Biodiversity Conservation Act 1999*. All are found on Aboriginal land, as well as conservation reserves and pastoral leases. The Victoria River nabarlek *P. c. concinna* has probably been extinct for some time [47,51]. Another species raised to the endangered category nationally is the Arnhem leaf-nosed bat *Hipposideros inornatus*, found almost exclusively on Aboriginal land. The fawn antechinus *Antechinus bellus*, found in far north NT, was listed as vulnerable nationally in 2015 (https://www.legislation.gov.au/Details/F2015L01912 (accessed on 23 April 2021)) and endangered in the NT in 2012. The spectacled flying-fox *Pteropus conspicillatus* was declared endangered in February 2019, due to significant population declines of more than 75% in just 15 years [52] and has been severely affected by extreme heat events, losing one third of its remaining population in November 2018.

Until recently, the populations of mammals on the large off-shore islands, such as Groote Eylandt (Anindilyakwa) and the Tiwi Islands (Bathurst and Melville) in the NT, were considered reasonably secure [23]. Alarmingly, recent studies have found declines of 60% to 90% of several species including the northern brown bandicoot *Isoodon macrourus*, black-footed tree-rat *Mesembriomys gouldii gouldii*, and brush-tailed rabbit-rat *Conilurus penicillatus* on Melville Island in only 15 years [53].

The status of the Queensland subspecies of black-footed tree-rat *M. g. rattoides* could not be determined due to the lack of studies [19,54]. The declines of northern brown bandicoots and black-footed tree-rats could not be attributed to any particular causes, therefore it is difficult to provide clear guidance for remedial management responses [53]. Brush-tailed rabbit-rat declines seemed to be associated with a combination of fire regimes, shrub cover, and feral cats [55].

In 2015, it was speculated that the Bramble Cay melomys *Melomys rubicola* (endemic to the Torres Strait) might be extinct [23,51], which has been confirmed by recent surveys [56,57], despite a recovery plan being in place [58]. The melomys was extirpated by sea-level rise associated with global warming, a major concern for the tropics [59,60].

Other species declines have been observed in Western Australia, although these observations are based on relatively short-term monitoring data [33,61,62], infrequent, dissimilar, and non-systematic surveys (e.g., [20,63]) and some only on inventory records and sub-fossil and fossil remains [64]. This is despite some areas, particularly the wetter north-western Kimberley region adjacent to the coast and on islands, showing reasonably healthy populations of many species [65,66]. One taxon, the once common Kimberley nabarlek, had not been recorded on the Kimberley mainland for over 40 years [66] until one was recorded by Indigenous rangers during surveys (T. Vigilante, A. Watson, pers. comm. 2019) (although they still occur on nearshore islands [65,67]). (Note: references to 'rangers' throughout the text are to Indigenous rangers employed mostly on Indigenous Protected Areas (IPAs)). These recent changes in threat status emphasize the concerns of experts and the Australian Government about their status (https://www.legislation.gov.au/Details/F2015L01912 (accessed on 23 April 2021)) and, critically, the absence of monitoring for most species.

**Table 1.** List of mammal species occurring in northern Australia at the time of European settlement (updated from Ziembicki et al. 2015) [23]. Note that this excludes the Wet Tropics area of northeastern Australia. Distribution (at the time of European settlement) is in addition to northern Australia: R = other parts of Australia; X = extralimital. Conservation status is given as for Australian national legislation (*Environment Protection and Biodiversity Conservation Act 1999*; EPBCA), the IUCN Red List, and as assessed in the *Action Plan for Australian Mammals 2012* (APAM) [51]. Conservation status categories: EX: extinct, EXW: extinct in the wild, CR(PE): critically endangered (possibly extinct), CR: critically endangered, EN: endangered, VU: vulnerable, NT (CD): near threatened (conservation dependent), NT: near threatened, LC: least concern, and DD: data deficient. Note that EPBCA and APAM status refer to Australian range only. Changes since the original table are highlighted in red font, with an arrow (>) to indicate changes; nl = not listed; 'n' = not listed as monitored (Woinarski et al. 2018b) [19]; for conservation status at the state/territory level, WA refers to listing in the Western Australian *Biodiversity Conservation Regulations 2018*, NT to listing in the Northern Territory's *Territory Parks and Wildlife Conservation Act 1976,* and Qld to listing in Queensland's *Nature Conservation Act 1992.* (1) [19]—max score 45 based on 9 evaluation metrics.

| Scientific Name | Common Name | Distribution | Conservation Status: Nationa or International Level | | | Conservation Status: State/Territory Level | | | Summed Monitoring Score (1) |
| --- | --- | --- | --- | --- | --- | --- | --- | --- | --- |
| | | | EPBCA | IUCN | APAM | WA | NT | Qld | Score + (Naïve Score) *** |
| TACHYGLOSSIDAE | | | | | | | | | |
| *Tachyglossus aculeatus* | Short-beaked echidna | RX | | LC | LC | | | SLC | 28 |
| *Zaglossus bruijnii* | Western long-beaked echidna | X | | CR | EX | | | | |
| ORNITHORHYNCHIDAE | | | | | | | | | |
| *Ornithorhynchus anatinus* | Platypus | R | | LC > NT | NT | | | | 25 |
| DASYURIDAE | | | | | | | | | |
| *Antechinomys laniger* | Kultarr | R | | LC | LC | | | LC | n |
| *Antechinus bellus* | Fawn antechinus | | VU | LC > VU | VU | | EN | | 19 |
| *Antechinus leo* | Cinnamon antechinus | | | LC | LC | | | LC | n |
| *Pseudantechinus bilarni* | Sandstone antechinus | | | NT > LC | LC | | | LC | 20 |
| *Pseudantechinus mimulus* | Carpentarian antechinus | | VU > nl | EN > NT | NT | | | LC | 15 |
| *Pseudantechinus ningbing* | Ningbing antechinus | | | LC | LC | | | | n |
| *Dasyurus hallucatus* | Northern quoll | R | EN | EN | EN | EN | CR | LC | 28 |
| *Dasyurus hallucatus hallucatus* | Northern quoll (NT) | | nl | | nl | | nl | | 28 ** |
| *Dasyurus hallucatus exilis* | Northern quoll (Kimberley, WA) | | nl | | nl | | nl | | n (28) |
| *Dasyurus hallucatus predator* | Northern quoll (Cape York, Qld) | | nl | | nl | | nl | | n (12) |
| *Dasyurus hallucatus (Pilbara)* | Northern quoll (Pilbara, WA) | | nl | | nl | | nl | | n (28) |
| *Dasyurus maculatus gracilis* | Spotted-tailed quoll (northern subspecies) | | | nl | EN | | | EN | 24 |

**Table 1.** *Cont.*

| Scientific Name | Common Name | Distribution | Conservation Status: Nationa or International Level | | | Conservation Status: State/Territory Level | | | Summed Monitoring Score (1) |
|---|---|---|---|---|---|---|---|---|---|
| | | | EPBCA | IUCN | APAM | WA | NT | Qld | Score + (Naïve Score) *** |
| TACHYGLOSSIDAE | | | | | | | | | |
| *Phascogale pirata* | Northern brush-tailed phascogale | | VU | VU | VU | | EN | | 9 |
| *Phascogale tapoatafa kimberleyensis* | Brush-tailed phascogale | R | | NT > nl | NT | EN | | | 0 |
| *Planigale ingrami* | Long-tailed planigale | R | | LC | LC | | | | n |
| *Planigale maculata* | Common planigale | R | | LC | LC | | | LC | n |
| *Sminthopsis archeri* | Chestnut dunnart | X | | DD | NT | | | NT | 0 |
| *Sminthopsis bindi* | Kakadu dunnart | | | LC > NT | NT | | | | 13 |
| *Sminthopsis butleri* | Butler's dunnart | | VU | VU | VU | EN > VU | NT > VU | | 14 |
| *Sminthopsis douglasi* | Julia Creek dunnart | | EN > VU | NT | NT | | | EN | 16 |
| *Sminthopsis macroura* | Stripe-faced dunnart | R | | LC | LC | | | LC | n |
| *Sminthopsis virginiae* | Red-cheeked dunnart | X | | LC | LC | | | | n |
| PERAMELIDAE | | | | | | | | | |
| *Echymipera rufescens* | Long-nosed echymipera | X | | LC | LC | | | LC | n |
| *Isoodon auratus auratus* | Golden bandicoot | R | VU * | VU > nl | VU | EN * > VU | EN | | 25 |
| *Isoodon macrourus* | Northern brown bandicoot | RX | | LC | LC | | | LC | n |
| *Isoodon peninsulae* | Cape York brown bandicoot | | | | LC | | | LC | n |
| *Perameles pallescens* | Northern Long-nosed bandicoot | | | | LC | | | LC | n |
| THYLACOMYIDAE | | | | | | | | | |
| *Macrotis lagotis* | Greater bilby | R | VU | VU | VU | EN > VU | VU | EN | 25 |
| PHASCOLARCTIDAE | | | | | | | | | |
| *Phascolarctos cinereus* | Koala | R | (VU) > EN | LC > VU | VU | | | VU | 28 |

**Table 1.** *Cont.*

| Scientific Name | Common Name | Distribution | Conservation Status: Nationa or International Level | | | Conservation Status: State/Territory Level | | | Summed Monitoring Score (1) |
|---|---|---|---|---|---|---|---|---|---|
| | | | EPBCA | IUCN | APAM | WA | NT | Qld | Score + (Naïve Score) *** |
| VOMBATIDAE | | | | | | | | | |
| *Lasiorhinus krefftii* | Northern hairy-nosed wombat | R | EN > CR | CR | CR | | | EN > CR | 41 |
| PETAURIDAE | | | | | | | | | |
| *Dactylopsila trivirgata* | Striped possum | RX | | LC | LC | | | LC | n |
| *Petaurus australis* | Yellow-bellied glider | R | | LC > NT | NT | | | | 24 |
| *Petaurus australis* unnamed subsp. | Yellow-bellied glider (northern subspecies) | | | | | | | EN | n |
| *Petaurus breviceps* | Sugar glider | RX | | LC | LC | | | LC | n |
| *Petaurus gracilis* | Mahogany glider | | EN | EN | EN | | | EN | 18 |
| *Petaurus norfolcensis* | Squirrel glider | R | | LC | LC | | | LC | n |
| PSEUDOCHEIRIDAE | | | | | | | | | |
| *Petauroides minor* | Northern greater glider | | | nl as sp. | | | | VU | n |
| *Petauroides volans* | Greater glider (southern) | R | VU | LC > VU | VU | | | VU | 19 |
| *Petropseudes dahli* | Rock ringtail possum | | | LC | LC | P3 | | LC | n |
| ACROBATIDAE | | | | | | | | | |
| *Acrobates pygmaeus* | Feathertail glider | R | | LC | LC | | | LC | n |
| PHALANGERIDAE | | | | | | | | | |
| *Spilocuscus maculatus* | Common spotted cuscus | RX | | LC | LC | | | LC | n |
| *Phalanger mimicus* | Southern common cuscus | RX | | LC | LC | | | LC | n |
| *Trichosurus vulpecula* | Common brushtail possum | R | | LC | LC | | | LC | n |
| *Trichosurus vulpecula arnhemensis* | Common brushtail possum (Arnhem subsp.) | | | nl | | | nl | | 16 |
| *Trichosurus vulpecula vulpecula* | Common brushtail possum (NT and WA) | | | nl | | | EN | | n |
| *Trichosurus vulpecula eburacensis* | Common brushtail possum (Cape York) | | | nl | | | | | n (0) |
| *Wyulda squamicaudata* | Scaly-tailed possum | | | DD > NT | NT | P4 (NT) | | | 24 |

**Table 1.** *Cont.*

| Scientific Name | Common Name | Distribution | Conservation Status: Nationa or International Level | | | Conservation Status: State/Territory Level | | | Summed Monitoring Score (1) |
|---|---|---|---|---|---|---|---|---|---|
| | | | EPBCA | IUCN | APAM | WA | NT | Qld | Score + (Naïve Score) *** |
| POTOROIDAE | | | | | | | | | |
| *Aepyprymnus rufescens* | Rufous bettong | R | | LC | LC | | | LC | n |
| *Bettongia lesueur* | Boodie | R | | NT | NT (CD) | | | | 33 |
| *Bettongia lesueur graii* | Boodie | R | | NT | NT (CD) | EN > EX | | EX | n |
| *Bettongia tropica* | Northern bettong | | EN | EN | EN | | | EN | 23 |
| MACROPODIDAE | | | | | | | | | |
| *Lagorchestes conspicillatus* | Spectacled hare-wallaby | RX | | LC | NT | | | LC | 0 |
| *Macropus agilis* | Agile wallaby | RX | | LC | LC | | | | n |
| *Osphranter (Macropus) antilopinus* | Antilopine wallaroo | | | LC | LC | | | | n |
| *Osphranter (Macropus) bernardus* | Black wallaroo | | | LC | NT | | | | 0 |
| *Notamacropus (Macropus) dorsalis* | Black-striped wallaby | R | | LC | LC | | | | |
| *Macropus giganteus* | Eastern grey kangaroo | R | | LC | LC | | | | |
| *Notamacropus (Macropus) parryi* | Whiptail wallaby | R | | LC | LC | | | | |
| *Macropus robustus* | Euro | R | | LC | LC | | | | |
| *Osphranter (Macropus) rufus* | Red kangaroo | R | | LC | LC | | | | |
| *Onychogalea fraenata* | Bridled nailtail wallaby | R | EN | EN > VU | VU | | | EN | 38 |
| *Onychogalea unguifera* | Northern nailtail wallaby | | | LC | LC | | | LC | n |
| *Petrogale assimilis* | Allied rock-wallaby | | | LC | LC | | | LC | n |
| *Petrogale brachyotis* | Western short-eared rock-wallaby | | | LC | LC | | | | 0 |
| *Petrogale burbidgei* | Monjon | | | NT | NT | P4 (NT) | | | 18 |
| *Petrogale coenensis* | Cape York rock-wallaby | | EN | NT > EN | EN | | | VU | 0 |

**Table 1.** *Cont.*

| Scientific Name | Common Name | Distribution | Conservation Status: Nationa or International Level | | | Conservation Status: State/Territory Level | | | Summed Monitoring Score (1) |
|---|---|---|---|---|---|---|---|---|---|
| | | | EPBCA | IUCN | APAM | WA | NT | Qld | Score + (Naïve Score) *** |
| MACROPODIDAE | | | | | | | | | |
| *Petrogale concinna* | Nabarlek | | | DD > EN | NT | | VU | | 14 |
| *Petrogale concinna canescens* | Nabarlek (Top End) | | VU > EN | nl | VU | | EN | | 0 |
| *Petrogale concinna concinna* | Nabarlek (Victoria River district) | | EN | nl | CR (poss. EX) | | CR (poss. EX) | | 0 |
| *Petrogale concinna monastria* | Kimberley nabarlek | | EN | nl | NT | EN | | | 15 |
| *Petrogale godmani* | Godman's rock-wallaby | | | LC > NT | NT | | | LC | 0 |
| *Petrogale herberti* | Herbert's rock-wallaby | | | LC | LC | | | LC | n |
| *Petrogale inornata* | Unadorned rock-wallaby | | | LC | LC | | | LC | n |
| *Petrogale lateralis* | Black-footed rock-wallaby | R | EN > nl | NT | EN | VU | | | 25 |
| *Petrogale lateralis kimberleyensis* | Black-footed rock-wallaby | R | EN | NT | EN | EN | | | 10 |
| *Petrogale mareeba* | Mareeba rock-wallaby | | | LC > NT | NT | | | LC | 0 |
| *Petrogale penicillata* | Brush-tailed rock-wallaby | | | VU | | | | VU | n |
| *Petrogale persephone* | Proserpine rock-wallaby | | EN | EN | EN | | | EN | 32 |
| *Petrogale purpureicollis* | Purple-necked rock-wallaby | | | LC > NT | NT | | | VU | 0 |
| *Petrogale sharmani* | Mount Claro rock-wallaby | | | NT > VU | VU | | | VU | 28 |
| *Petrogale wilkinsi* | Eastern short-eared rock-wallaby | | | LC > nl | LC | | | LC | n |
| *Thylogale stigmatica* | Red-legged pademelon | RX | | LC | LC | | | LC | n |
| *Wallabia bicolor* | Swamp wallaby | R | | LC | LC | | | LC | n |

**Table 1.** *Cont.*

| Scientific Name | Common Name | Distribution | Conservation Status: Nationa or International Level | | | Conservation Status: State/Territory Level | | | Summed Monitoring Score (1) |
| --- | --- | --- | --- | --- | --- | --- | --- | --- | --- |
| | | | EPBCA | IUCN | APAM | WA | NT | Qld | Score + (Naïve Score) *** |
| NOTORYCTIDAE | | | | | | | | | |
| *Notoryctes caurinus* | Kakarratul | R | EN > nl | DD > LC | LC | EN > P4 (NT) | | | n (0) |
| PTEROPODIDAE | | | | | | | | | |
| *Dobsonia magna* | Bare-backed fruit bat | X | | LC | LC | | | LC | n |
| *Macroglossus minimus* | Northern blossom bat | RX | | LC | LC | | | LC | n |
| *Nyctimene robinsoni* | Eastern tube-nosed bat | R | | LC | LC | | | LC | n |
| *Pteropus alecto* | Black flying-fox | RX | | LC | LC | | | LC | n |
| *Pteropus conspicillatus* | Spectacled flying-fox | RX | VU > EN | LC > EN | NT (CD) | | | EN | 28 |
| *Pteropus macrotis* | Large-eared flying-fox | X | | LC | LC | | | | n |
| *Pteropus scapulatus* | Little red flying-fox | R | | LC | LC | | | LC | n |
| *Syconycteris australis* | Eastern blossom bat | RX | | LC | LC | | | LC | n |
| MEGADERMATIDAE | | | | | | | | | |
| *Macroderma gigas* | Ghost bat | R | VU | VU | VU | NT > VU | NT | EN | 19 |
| RHINOLOPHIDAE | | | | | | | | | |
| *Rhinolophus megaphyllus* | Eastern horseshoe-bat | RX | | LC | LC | | | | n |
| *Rhinolophus 'intermediate'* | Lesser large-eared horseshoe-bat | | | | VU | | | | 0 |
| *Rhinolophus philippinensis (robertsi)* | Greater large-eared horseshoe-bat | | VU | VU >LC | NT | | | EN | 0 |
| HIPPOSIDERIDAE | | | | | | | | | |
| *Hipposideros ater* | Dusky leaf-nosed bat | X | | LC | LC | | | | n |
| *Hipposideros cervinus* | Fawn leaf-nosed Nat | X | | LC | NT | | | VU | 9 |
| *Hipposideros diadema reginae* | Diadem leaf-nosed bat | X | | LC (as *H. diadema*) | NT | | | VU | 0 |
| *Hipposideros inornatus* | Arnhem leaf-nosed bat | | EN | VU | EN | | VU | | 0 |
| *Hipposideros semoni* | Semon's leaf-nosed bat | RX | EN > VU | DD > LC | NT | | | EN | 13 |

**Table 1.** *Cont.*

| Scientific Name | Common Name | Distribution | Conservation Status: Nationa or International Level | | | Conservation Status: State/Territory Level | | | Summed Monitoring Score (1) |
|---|---|---|---|---|---|---|---|---|---|
| | | | EPBCA | IUCN | APAM | WA | NT | Qld | Score + (Naïve Score) *** |
| HIPPOSIDERIDAE | | | | | | | | | |
| *Hipposideros stenotis* | Northern leaf-nosed bat | | | LC > VU | NT | NT > P2 | | VU | 0 |
| *Rhinonicteris aurantia* | Orange leaf-nosed bat | R | | LC | LC | EN > P4 (NT) | | VU | n |
| EMBALLONURIDAE | | | | | | | | | |
| *Saccolaimus flaviventris* | Yellow-bellied sheath-tailed bat | R | | LC | LC | | | LC | n |
| *Saccolaimus mixtus* | Cape York sheath-tailed bat | X | | DD > NT | NT | | | LC | 0 |
| *Saccolaimus saccolaimus nudicluniatus* | Bare-rumped sheath-tailed bat | X | | LC | NT | | NT | EN | 0 |
| *Taphozous australis* | Coastal sheath-tailed bat | RX | | NT | NT | | | NT | 0 |
| *Taphozous georgianus* | Common sheath-tailed bat | R | | LC | LC | | | LC | n |
| *Taphozous kapalgensis* | Arnhem sheath-tailed bat | | | LC | LC | | | | n |
| *Taphazous troughtoni* | Troughton's sheath-tailed bat | | | DD > nl | LC | | | LC | n |
| MOLOSSIDAE | | | | | | | | | |
| *Austronomus australis* | White-striped free-tailed bat | R | | LC | LC | | | LC | n |
| *Chaerephon jobensis* | Greater northern free-tailed bat | RX | | LC | LC | | | LC | n |
| *Setirostris (Mormopterus) eleryi* | Bristle-faced free-tailed bat | R | | LC | LC | | | LC | 15 |
| *Ozimops (Mormopterus) lumsdenae* | Northern free-tailed bat | R | | LC | | | | LC | n |
| *Mormopterus ridei* | Eastern free-tailed bat | R | | LC | | | | LC | n |
| *Ozimops (Mormopterus) halli* | Cape York free-tailed bat | | | DD | | | | LC | 0 |
| *Ozimops (Mormopterus) cobourgianus* | North-western free-tailed bat | R | | LC | | | | | n |
| MINIOPTERIDAE | | | | | | | | | |
| *Miniopterus australis* | Little bent-winged bat | RX | | LC | LC | | | LC | |
| *Miniopterus orianae* | Common bent-wing bat | RX | | NT > nl | LC | | | | 36 |

**Table 1.** *Cont.*

| Scientific Name | Common Name | Distribution | Conservation Status: Nationa or International Level | | | Conservation Status: State/Territory Level | | | Summed Monitoring Score (1) |
|---|---|---|---|---|---|---|---|---|---|
| | | | EPBCA | IUCN | APAM | WA | NT | Qld | Score + (Naïve Score) *** |
| VESPERTILIONIDAE | | | | | | | | | |
| *Chalinolobus gouldii* | Gould's wattled bat | R | | LC | LC | | | LC | n |
| *Chalinolobus nigrogriseus* | Hoary wattled bat | RX | | LC | LC | | | LC | n |
| *Murina florium* | Flute-nosed bat | RX | | LC | NT | | | VU | 13 |
| *Myotis macropus* | Large-footed myotis | RX | | LC | LC | | | LC | n |
| *Nyctophilus arnhemensis* | Northern long-eared bat | R | | LC | LC | | | LC | n |
| *Nyctophilus bifax* | Eastern long-eared bat | RX | | LC | LC | | | LC | n |
| *Nyctophilus daedalus* | Pallid long-eared bat | | | LC | | | | LC | n |
| *Nyctophilus geoffroyi* | Lesser long-eared bat | R | | LC | LC | | | LC | n |
| *Nyctophilus gouldi* | Gould's long-eared bat | R | | LC | LC | | | LC | n |
| *Nyctophilus walkeri* | Pygmy long-eared bat | | | LC | LC | | | LC | n |
| *Phoniscus papuensis* | Golden-tipped bat | RX | | LC | LC | | | LC | n |
| *Pipistrellus adamsi* | Cape York pipistrelle | | | LC | LC | | | LC | n |
| *Pipistrellus westralis* | Northern pipistrelle | | | LC | LC | | | LC | n |
| *Scotorepens balstoni* | Inland broad-nosed bat | R | | LC | LC | | | LC | n |
| *Scotorepens greyii* | Little broad-nosed bat | R | | LC | LC | | | LC | n |
| *Scotorepens sanborni* | Northern broad-nosed bat | RX | | LC | LC | | | LC | n |
| *Vespadelus caurinus* | Western cave-bat | | | LC | LC | | | LC | n |
| *Vespadelus douglasorum* | Yellow-lipped cave bat | | | LC | LC | NT > P2 | | | n |
| *Vespadelus finlaysoni* | Inland cave bat | R | | LC | LC | | | LC | n |
| *Vespadelus troughtoni* | Eastern cave bat | R | | LC | LC | | | LC | n |
| MURIDAE | | | | | | | | | |
| *Conilurus capricornensis* | Capricornian rabbit-rat | R | | EX | EX | | | | n |
| *Conilurus penicillatus* | Brush-tailed rabbit-rat | X | VU | NT > VU | VU | | EN | VU | 13 |
| *Conilurus penicillatus melibius* | Brush-tailed rabbit-rat (Tiwi Islands) | | nl | nl | VU | | | | 24 |

Table 1. *Cont.*

| Scientific Name | Common Name | Distribution | Conservation Status: Nationa or International Level | | | Conservation Status: State/Territory Level | | | Summed Monitoring Score (1) |
|---|---|---|---|---|---|---|---|---|---|
| | | | EPBCA | IUCN | APAM | WA | NT | Qld | Score + (Naïve Score) *** |
| MURIDAE | | | | | | | | | |
| *Conilurus penicillatus penicillatus* | Brush-tailed rabbit-rat (Kimberley, Top End) | | nl | nl | VU | | VU | | 18 |
| *Hydromys chrysogaster* | Water-rat | RX | | LC | LC | NT | | LC | n |
| *Leggadina lakedownensis* | Northern short-tailed mouse | R | | LC | LC | NT | | LC | n |
| *Melomys burtoni* | Grassland melomys | R | | LC | LC | | | LC | n |
| *Melomys capensis* | Cape York melomys | | | LC | LC | | | LC | n |
| *Melomys cervinipes* | Fawn-footed melomys | R | | LC | LC | | | LC | n |
| *Melomys rubicola* | Bramble Cay melomys | | EN > EX | CR > EX | CR (PE) | | EXW | | n |
| *Mesembriomys gouldii* | Black-footed tree-rat | | | NT > VU | VU | | | | 16 |
| *Mesembriomys gouldii gouldii* (Kimberley and mainland NT) | Black-footed tree-rat | | nl > EN | nl | VU | | EN | LC | 16 |
| *Mesembriomys gouldii melvillensis* (Melville Island) | Black-footed tree-rat | | nl > VU | nl | VU | | VU | LC | 22 |
| *Mesembriomys gouldii rattoides* (north Queensland) | Black-footed tree-rat | | nl > VU | nl | VU | | | LC | 0 |
| *Mesembriomys macrurus* | Golden-backed tree-rat | | VU > nl | LC > NT | NT | P4 (NT) | CR | | 24 |
| *Notomys alexis* | Spinifex hopping-mouse | R | | LC | LC | | | LC | n |
| *Notomys aquilo* | Northern hopping-mouse | VU | VU | EN | VU | | VU | VU | 16 |
| *Notomys fuscus* | Dusky hopping-mouse | | | nl > VU | | | | EN | 20 |
| *Pogonomys* sp. | Tree mouse | R | | LC | LC | | | | n |
| *Pseudomys calabyi* | Kakadu pebble-mouse | | | VU | NT | | NT | | 18 |
| *Pseudomys delicatulus* | Delicate mouse | RX | | LC | LC | | | LC | n |
| *Pseudomys desertor* | Desert mouse | R | | LC | LC | | | LC | n |
| *Pseudomys gracilicaudatus* | Eastern chestnut mouse | R | | LC | LC | | | LC | n |
| *Pseudomys johnsoni* | Central pebble-mouse | R | | LC | LC | | | LC | n |
| *Pseudomys nanus* | Western chestnut Mouse | | | LC | LC | | | LC | n |

**Table 1.** *Cont.*

| Scientific Name | Common Name | Distribution | Conservation Status: Nationa or International Level | | | Conservation Status: State/Territory Level | | | Summed Monitoring Score (1) |
|---|---|---|---|---|---|---|---|---|---|
| | | | EPBCA | IUCN | APAM | WA | NT | Qld | Score + (Naïve Score) *** |
| MURIDAE | | | | | | | | | |
| *Pseudomys patrius* | Eastern pebble-mouse | R | | LC | LC | | | LC | n |
| *Rattus colletti* | Dusky rat | | | LC | LC | | | LC | n |
| *Rattus fuscipes* | Bush rat | R | | LC | LC | | | LC | n |
| *Rattus leucopus* | Cape York rat | RX | | LC | LC | | | LC | n |
| *Rattus lutreolus* | Swamp rat | R | | LC | LC | | | LC | n |
| *Rattus sordidus* | Canefield rat | R | | LC | LC | | CR > nl | LC | n |
| *Rattus tunneyi* | Pale field-rat | R | | LC | LC | | VU | LC | 18 |
| *Rattus villosissimus* | Long-haired rat | R | | LC | LC | | | LC | n |
| *Uromys caudimaculatus* | Giant white-tailed rat | RX | | LC | LC | | | LC | n |
| *Xeromys myoides* | Water mouse | RX | VU | VU | VU | | DD | VU | 21 |
| *Zyzomys argurus* | Common rock-rat | R | | LC | LC | | | LC | n |
| *Zyzomys maini* | Arnhem rock-rat | VU | VU | NT > VU | VU | | VU | | 25 |
| *Zyzomys palatalis* | Carpentarian rock-rat | EN | EN | CR | CR | | CR > EN | | 24 |
| *Zyzomys woodwardi* | Kimberley rock-rat | LC | | LC | | | | | n |
| CANIDAE | | | | | | | | | |
| *Canis dingo* | Dingo | R | | LC | NT | | | | 17 |

* for subspecies *Isoodon auratus auratus* and *I. a. barrowensis*; ** assumed assessment population from published information; *** score from Woinarski et al. 2018b [19], naïve score based on limited recent knowledge; nl = not listed.

### 2.3. Taxonomic Revisions and Imminent Extinctions

More species are likely to be described in north Australia, and the taxonomy of a number of species need to be resolved [23]. Since the Ziembicki et al. (2015) [23] review, there have been revisions of two major taxa, one within the order Peramelemorphia [68] and one in the genus *Dasyurus* [69] (see below). Disturbingly, the revision of the Peramelemorphia (bandicoots and bilbies) prompted Travouillon and Phillips (2018) [68] to suggest that the known number of recently extinct bandicoots is likely to increase.

The extinctions of fauna are difficult to predict, but a recent expert elicitation on Australian birds and mammals produced alarming results [47,70]. The mammals and birds that were previously considered to be at most risk [51,71], were analyzed further in order to forecast the extinction risk and it was found that of the 20 mammal species most likely to go extinct in the next 20 years, almost half (9) occur in north Australia [47] (Table 2). (This account overlooked the spectacled flying-fox *Pteropus conspicillatus* [52,72], which takes the total in north Australia to 10). The extant ranges of these species occur on Aboriginal land, pastoral lands, and national parks. The endangered birds fared better in north Australia, with only one considered to be in danger of imminent extinction [47]. These listings are based on thorough assessments of all Australian birds and mammals, and have examined only the most highly ranked species in terms of most threatened status as follows: 40 birds, and 41 mammals [47].

**Table 2.** The likelihood of extinction in the next 20 years for *northern* mammals and birds considered most imperiled (extract from Table 1 of [47]).

| Rank (Out of 20 for Each Taxon) | Mammals | Extinction Likelihood |
|:---:|:---:|:---:|
| 2 | Northern hopping-mouse, *Notomys aquilo* | 0.48 |
| 3 | Carpentarian rock-rat, *Zyzomys palatalis* | 0.44 |
| 5 | Black-footed tree-rat (Kimberley and mainland NT), *Mesembriomys gouldii gouldii* | 0.39 |
| 8 | Nabarlek (Top End), *Petrogale concinna canescens* | 0.29 |
| 9 | Brush-tailed phascogale (Kimberley), *Phascogale tapoatafa kimberleyensis* | 0.28 |
| 10 | Brush-tailed rabbit-rat (Kimberley and Top End), *Conilurus penicillatus penicillatus* | 0.25 |
| 12 | Northern brush-tailed phascogale, *Phascogale pirata* | 0.23 |
| 15 | Brush-tailed rabbit-rat (Tiwi Islands), *Conilurus penicillatus melibius* | 0.21 |
| 19 | Northern bettong, *Bettongia tropica* | 0.14 |
| | **Birds** | |
| 20 | Alligator Rivers yellow chat, *Epthianura crocea tunneyi* | 0.15 |

The forecast rates of extinction are about five times higher than what has been occurring since European settlement, now at one to two species per decade [73], and is ~1000 times the background rate [47]. This increased rate is predicated on the basis that the intensity of many threats, including climate change, will increase and further extinctions are likely unless management efforts are increased substantially [47]. The need to implement responses and management actions raises the complex issue, however, of what management actions are appropriate when studies cited previously have shown contradictory results from management actions, such as prescribed fire and removal of large exotic herbivores, combined with the lack of monitoring of most species and their distinct populations.

### 2.4. Northern Quoll as an Important Conservation Case Study

Taxa within the genus *Dasyurus* have been revised to reveal four genetically divergent subspecies of northern quoll *Dasyurus hallucatus* [69,74] with significant biogeographic gaps between the populations [69] (Figure 1). These lineages may have diverged two to five million years ago (preliminary analysis by M. Westerman, pers. comm.), which means that they have evolved into new geographic forms and may qualify as subspecies [69]. From a conservation perspective, this is important in that most of the published studies of northern

quoll status have been limited to the NT quolls (subspecies *hallucatus*). The northern quoll's status varies from critically endangered in the NT, to endangered in WA and Australia-wide, and least concern in Qld, but the legislation recognizes only one species, and no subspecies [75], essentially ignoring the biogeographic distinctiveness of the subspecies and their individual statuses.

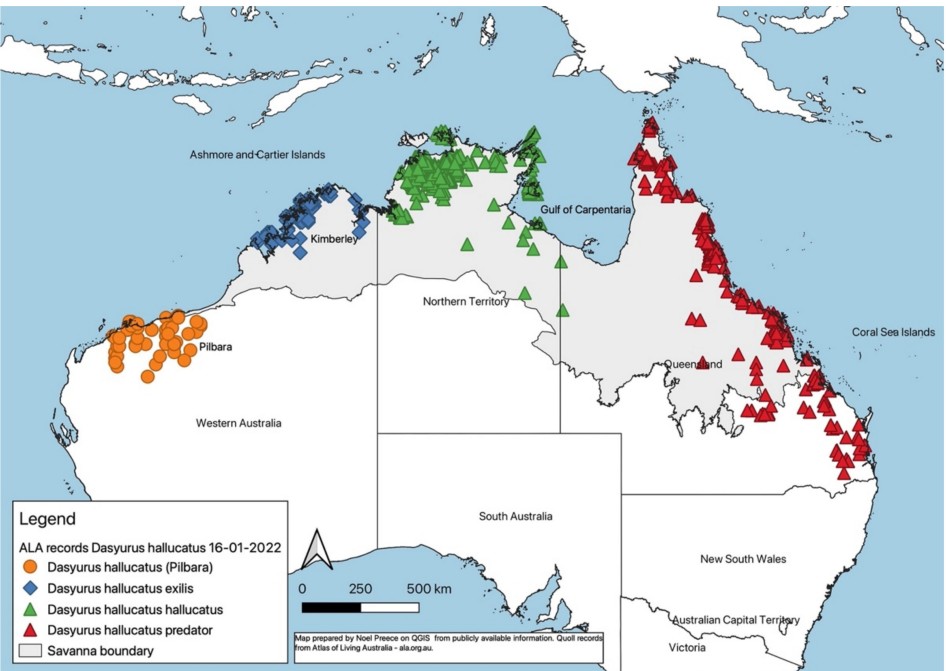

**Figure 1.** Records of northern quolls *Dasyurus hallucatus* from the Atlas of Living Australia (ALA) (https://www.ala.org.au/ (accessed on 16 January 2022)). Note the large gaps in distribution across the southern Gulf of Carpentaria and the northern and western distributions, and the minor gap between the Northern Territory (*D. h. hallucatus*) and Western Australian (*D. h. exilis*) subspecies.

Northern quoll distributions and abundance have declined significantly across the NT [66,73], and were detected in standard fauna surveys since at least the early 1990s when the numbers trapped were high. Fish River Station, owned by the Indigenous Land and Sea Corporation, has a small population recorded at only one site and concern about their survival was expressed by researchers and Indigenous custodians [76]. There are some healthy populations on Groote Eylandt in the Gulf of Carpentaria (G. Gillespie, pers. comm. 2018; [77]), but this is isolated from the mainland and is free of cane toads *Rhinella marina*, which are lethally toxic to quolls [78]. Anindilyakwa Land Council runs a biosecurity program to prevent the toads from establishing on Groote Eylandt [79] and so far this has been successful.

Although [19] suggest that there is moderate monitoring for the northern quoll, most subspecies are either not systematically monitored or have limited monitoring in parts of their range beyond the NT [80]. In Western Australia, a Pilbara population is monitored as part of mining operations [81] and seems to maintain a healthy population [82]; recent research has improved knowledge of this population's spatial needs [83,84]. In the Kimberley area, some studies have been conducted [85] in the Mitchell Plateau region where populations also seemed stable, although the incursion of cane toad in recent years may detrimentally affect this population [78]. In north Queensland, populations of subspecies *D. h. predator* are known from the Mt Emerald wind farm near Atherton [86], from the Black Mountain area near Cooktown [87], but not from near Weipa on the northern Cape York Peninsula where it has been recorded previously [88]. The monitoring at the wind farm has shown that the population is relatively stable at several sites [86]. None of the other populations are monitored, so there are no data on their trends.

Concern about the severe declines of species, such as the northern quoll, have led to translocations and reintroductions for their ultimate survival [89–93]. For example, a recent study using northern quolls sourced from Queensland and the NT to test out-breeding depression found, with reservations, that targeted gene flow could be used in some situations to help recover the species [93] but failed to address the maintenance of distinct genetic lineages wherein reintroducing species from one subspecies population to another could be contrary to conservation objectives (Convention on Biological Diversity's Aichi targets). Another study of northern quolls translocated to predator-free islands in the NT proved successful demographically [92], but a subsequent study found that the island quolls, which had been isolated from predators for only 13 generations, showed no recognition of nor aversion to predators (such as feral cats and dingoes) so reintroduction of the island quolls to the mainland would be problematic [94]. The latter study addressed the translocation criteria established by IUCN [95] but failed to address genetic lineages explicitly (e.g., [92]). We argue that until the taxonomic distinctiveness of species separated geographically for many generations, such as the northern quoll, are resolved, caution about translocations and reintroductions must be exercised and addressed explicitly in proposals and they should comply with the IUCN/SSC [95] guidelines.

## 3. Causes: Cattle, Cats, Climate Change, Cane Toads, Diseases, and Fire?

There is an on-going debate about how and in what combinations feral predators, grazing by introduced herbivores, cane toads, diseases, and changed fire regimes are factors in the declines of different species [22,30,43,44,96–100]. Modelling of the causes and associations of the declines of mammals has been limited [23] but clearly there are multiple stressors [101]. It had been assumed, until recently, that mammal declines had not occurred in Arnhem Land in the NT [102] and on some of the larger islands of the tropical savannas [53,65] but follow-up monitoring has shown that some severe declines have occurred in areas where fire management practices had been considered the most appropriate for biodiversity conservation [29,53].

We explore these possible causes in more detail, noting that all of the uncertainties around the causes and effects require systematic monitoring in order to resolve the contradictory evidence and to examine the outcomes of the management actions.

### 3.1. Resurgence of Fire Application

Fire is one of the main drivers of vegetation structure and composition in the tropical savanna region and many studies have demonstrated a negative relationship between high fire frequency, extent and intensity, and the richness and abundance of terrestrial fauna in north Australian savanna landscapes [14,23,29,103–106], but the absence of fire can also be a driver of declines through lack of rejuvenation of plant species [107]. The season, patchiness, and size of the burnt areas are also factors in the heterogeneity of the post-fire recovery landscape [108].

In efforts to reduce the large-scale wildfires that have occurred in recent decades and reduce greenhouse gas emissions, savanna burning projects that resurrect the application of a more patchy and variable fire regime have been implemented [108,109]. The traditional fire management practices created a fine-scale mosaic of various post-fire vegetation ages, but broke down through the massive disenfranchisement of Indigenous people from their land [110–113] and there has been a strong push to re-deploy customary fire knowledge to fire management across the savannas, and to reconstruct customary relationships, local languages, protocols, and means for cooperation [108,114]. This push has found support in the Emissions Reduction Fund (ERF) Savanna Fire Management Determination, which was established in 2012 [115]. Overall, the program has been a success on pastoral, conservation, and Indigenous lands, with 25% of the higher-rainfall (>600 mm/a) savanna region (1.2 M km$^2$) now under this program and showing substantial improvements in fire regimes, including a reduction in wildfires [14].

Engaging Indigenous people to re-build modern forms of savanna burning practices has resulted in the re-application of traditional burning and modified methods influenced by western-style fire authorities and legislated dictates and constraints [111] including long linear fire lines, often lit along roads, that create continuous fire fronts with little regard for patchiness (J. Russell-Smith pers. comm. 2019). Nevertheless, re-introduced savanna burning has shown promise for managing novel problems, such as increased emissions from wildfires, invasive species, and climate-induced increases in fire size [116] and non-Indigenous fire practitioners have gained invaluable knowledge about how to apply fire to the landscape [108,117]. Healthy Country Plans contribute to the effort to reintroduce fire for cultural and biodiversity values [118–122]. But this renewed interest in applying 'traditional' ways of burning and in prescribed fire is influenced by the need to conform with regulations on the timing of burning in order to gain financial returns from emissions reductions from the ERF, and perhaps less about improving biodiversity benefits [123] or re-invigorating traditional fire management practices [124].

The savanna burning method establishes that the early dry season (EDS) finishes on 31 July and late dry season (LDS) starts on 1 August each year for the whole of the high rainfall savanna region. These seasons are based on the behavior of fire from research in the NT that shows that fires self-extinguish overnight before the end of July [125]. However, the savanna burning method has created a binary partition of early dry and late dry season burning [126], which leaves little room for the application of traditional Indigenous burning, such as burning throughout the year, and according to the curing condition of grasses and other factors [14,24,110,117,123,127,128]. It also does not account for dry seasons starting later in the east of the region and annual variations among El Niño and La Niña years, leaving little time in some areas to complete burns before the cut-off date (Figure 2). There is a wide variation across the savanna region of the timing and trends of seasons, the effects of seasonal conditions on fire behavior, and responses that need to be considered [129]. The current practices need to be reconsidered for application across the savannas in accordance with regional variation [14], but changing the cut-off date for EDS is a major challenge that requires new techniques in remote sensing in order to determine the fire severity [126].

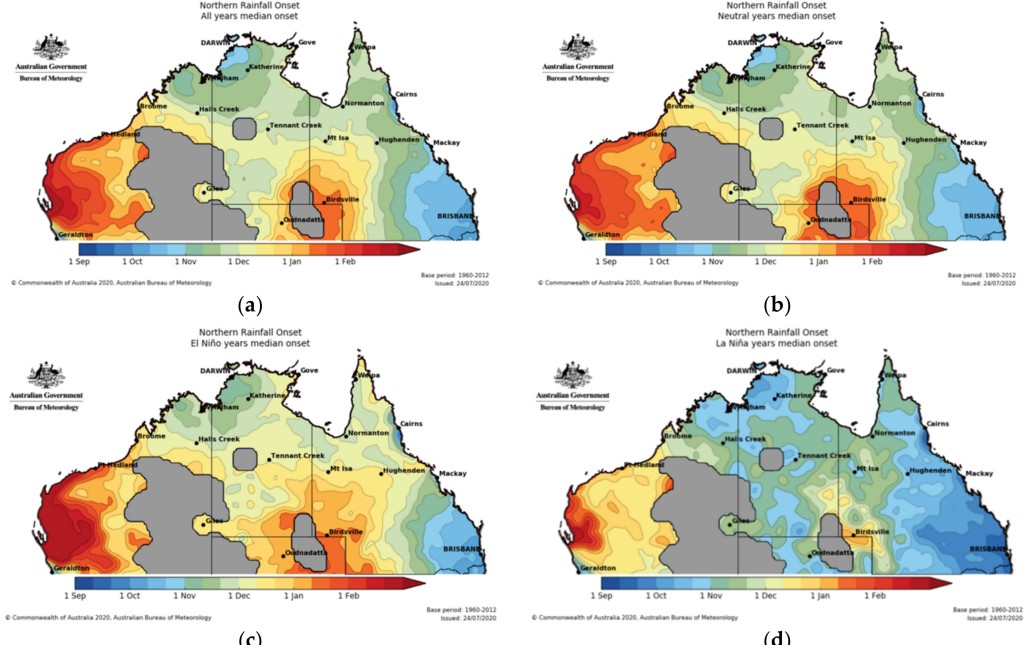

**Figure 2.** Northern rainfall onset medians showing differences in timing and trends of seasons: (**a**) All years median onset, (**b**) Neutral years median onset, (**c**) El Niño years median onset, and (**d**) La Niña year median onset [130] (accessed on 10 December 2021).

*3.2. Unsupported Assumptions Related to Fire Management*

Part of the problem of determining tangible outcomes from savanna burning is that few have been measured explicitly [24,32,131], and unsupported assumptions about the benefits of particular fire regimes for biodiversity are made, even today [109]. Monitoring programs designed to examine these outcomes are needed in order to provide empirical evidence and theoretical underpinning of the benefits of changing fire frequency and activity from late to early season fires [24,123,129]. The paucity of empirical evidence exists despite plans to benchmark biodiversity by identifying biodiversity indicators, condition targets, and associated assessment regimes [114]. The general rules about prescribed burning do not account for a number of factors, including individual species responses, and assumptions that prescribed burning will force vegetation into a somewhat nebulous 'natural state' [132]. Importantly, a single or a few fires do not necessarily change vegetation types to a new state, as vegetation dynamics operate at decadal scales [123,133] not annual or multi-annual fire scales. The paradigms have also been criticized because it is unclear which particular aspects of pyrodiversity should be manipulated for the best outcomes—frequency, intensity, season, patch size, heterogeneity, age class, or a combination of some or all of these [131,134]. While early dry season fire frequency is most likely influencing vegetation and fauna patterns, more subtle metrics are required in order to account for the responses of biodiversity to fire regimes [123].

High fire frequency has been positively associated with mammal declines (e.g., see references in [135]), but more recent studies in Cape York Peninsula have indicated that fire frequency, in both the early and late dry seasons, influences only some components of terrestrial fauna [123]. Increasing early dry-season fire frequency causes a complex biodiversity response and may have a slightly negative effect on mammal richness and abundance [123]. It has also been demonstrated that fire extent, which is a combination of fire size and fire frequency, is most important in the conservation of small mammals and more important than the proportion of the surrounding area burnt, fire patchiness, and point-based fire frequency [98].

Fire frequency has been found to be a significant, but not consistently positive or negative, predictor of abundance of some species, but not for others, on Melville Island in the NT [100,136] reinforcing the finding that a fire regime that supports biodiversity in one system does not necessarily support it in another [137]. Microbats from different foraging guilds showed variable responses to fire intensity on Cape York Peninsula [138]. The pale field-rat in the Kimberley region showed strong effects of size and spatial pattern of fires, with high mortality associated with more complete burns [139]. Recovery was associated with in situ survivors within unburnt refuges after patchy fires, and recolonization from areas outside of the burnt areas [139]. Some specialist rodents and large marsupials showed a positive response to early dry season burning, while generalist rodents showed a negative response in the western Kimberley region [140]. Although the results were not conclusive, Radford et al. [140] found that two arboreal rodents, the brush-tailed rabbit-rat and the golden-backed tree-rat *Mesembriomys macrurus,* responded positively to the application of prescribed fire in the form of patch or mosaic burning in the Kimberley region, which is contrary to the findings of the detrimental effects on both species in the NT [51,141,142]. These new results reveal a more complex picture of mammal fire responses than previously realized in the north Australian contexts [140].

Vertebrate diversity and abundance are promoted by natural heterogeneity in the savanna ecosystem landscape [123] and landscape heterogeneity can influence, and species distributions and abundances can be influenced by, fire regimes [143]. Burning areas that naturally experience more frequent fires might be more effective in protecting fire exclusion or infrequent fire areas [132]. The exclusion of fires is a difficult task across the savanna region, and even in carefully managed exclusion areas that have been protected from fire for more than 20 years, even short-term reintroduction of medium intensity fires has been found to revert the vegetation from a woodland to its pre-existing grassy savanna state [144].

Multiple authors have recommended that a priority conservation action was intensive fire management aimed at increasing the extent of longer-unburnt habitat and in delivering fine-scale patch burning [22,23,145]. A mosaic of fire patches of different ages also has been considered to be the best option for a range of birds, mammals, amphibians, and reptiles found in the savanna landscapes [146], but the mosaic hypothesis is not sufficiently nuanced for management to be effective [136]. Non-random targeted patch burning could be a more realistic goal than that suggested in Ziembicki et al. [23] of creating large long-unburnt areas [132], based on findings that the likelihood of long-unburnt habitat patches surviving in landscapes that are dominated by large fires is extremely low [116]. Creating many small fire patches by prescribed burning in more traditional ways creates patches of older vegetation, even though the total area burnt may be equivalent to a non-anthropogenic wildfire regime [116]. It is unrealistic, however, to expect a consistent predictable response from fauna across all landscapes at moderate spatial and temporal scales [100,132]. Fire regimes that optimize habitat resources for recruitment may be required and might be achieved by a reduction in fire frequency and managing fuel loads in order to prevent an increase in fire intensity [147]. While there are so many uncertainties, concerted fire management might best prioritize areas of highest value for biodiversity, whether they be areas rich in fauna and flora, or areas where the more valued species occur, an aim of traditional burning expressed by Aboriginal researchers on western Cape York [117]. The assessment of a biodiversity benefit from a particular fire management strategy requires on-ground measurement of responses from each targeted taxon in different locations and over long time periods [132].

### 3.3. Introduced Animals

Surprisingly, the relationship between introduced fauna and native faunal declines and extinctions in northern Australia is not well known [148] as studies show contradictory results of management, such as in the Kimberley region, where some fauna recovered after cattle removal [33], contrasting with a study in the Einasleigh Uplands that showed both upward and downward trends [40]. But most studies show a negative relationship, with more intensive cattle grazing being associated with lower richness and abundance of native fauna, particularly mammals [43,107,149]. The declines of mammals may be linked to the increasing abundance of introduced grazing animals and intensification of grazing across the landscape [150]. Recent evidence on grazing impacts [62,97,151–154] suggests that, for some species, any level of grazing by introduced herbivores may be detrimental.

Feral predators, such as cats, and predation on the introduced cane toad [23,43,44,155,156], have been implicated as some of the primary causes of declines of mammals and other fauna in the savanna region. Few predators can survive the effects of cane toad toxins if ingested [157–159] but these direct effects are confined to some reptile and mammalian predators that consume them. A recent study in the Kimberley and NT regions hypothesizes that cane toad invasion may have triggered reductions in apex predators that lead to greater impacts on fauna from meso-predators [99].

There are estimated to be somewhere around 1.4 to 3.4 million feral cats across Australia, around one feral cat per four square kilometers [160]. There is no doubt that feral cats eat large numbers of native fauna [161–163] but the studies are not conclusive on the actual effects on prey populations generally [162–164], and observed effects do not necessarily equate to causation [160]. It has been argued that feral cats have led to the demise of mammalian fauna [23,29,43,44,156,165], and an exclusion study in the Kimberley region suggested that feral cats can extirpate local populations of native mammals [35]. Whether or not feral cats can cause extinctions generally in open landscapes is still not clear. A study in the NT that translocated native long-haired rats, *Rattus villosissimus,* to experimental compounds and found that cats extirpated the rats within the compounds and prompted the speculation that feral cats can send native mammals to extinction 'in a continental setting' [156]. The study was confounded, however, as the rats were sourced from a cat-free island, they were captive-bred [23], the enclosures acted as islands where

the rats had no escape options, and there was no account made of other potential rat predators (raptors, snakes). As this species is known to reproduce vigorously in the right conditions [166,167], the subsequent death of the cat-free experimental animals suggests that conditions for population survival were not ideal, or that predation by non-feline predators was strong [35].

While it has been shown that reptile fauna can respond to the removal of cats from cat-proof enclosures [168], this response was confounded with the effects of fire regimes. There was no response over two years by mammals to cat exclusion, but counter-intuitively, northern quolls increased in adjacent areas where feral cats had not been controlled [169]. Cats and dingoes can have a negative effect, as was found on Mornington Sanctuary in the Kimberley region, where the mortality of rats and mice increased after low intensity and high intensity fire treatments, and there was direct evidence of predation by cats on native rodents [34].

Some studies [25,170,171] suggest that dingoes (a native canid; [172]), which are common in the region, have a role in controlling feral cats, and that dingoes are less of a threat to native fauna [61], although this relationship is also not clear [173]. Recent analyses of feral cat and dingo feces from Kakadu National Park have shown that dingoes consume a wide range of native mammals, and that they, coupled with feral cats, could have an impact on the small to medium mammals, especially in habitats disturbed by fire or grazing by introduced cattle [174]. Cats have been implicated in the near-extinction of the brush-tailed rabbit-rat on Melville Island off the NT coast, where populations are surviving in areas with high shrub density where feral cats are in low density, and the management and retention of areas with high shrub density is considered vital for its survival [55].

Few studies have been undertaken to test the impacts of feral predators on fauna, but those that have point mostly to impacts on contained or restricted populations (either on islands or in fenced exclosures) [175]. Other studies have suggested that predators alone are not the cause of mainland extinctions [20,165], but are linked to habitat changes, especially the loss of ground cover, making prey mammals more vulnerable to predation by cats and dingoes [43,97,98,170,174,176].

The first Threatened Species Strategy of the Australian Government [177] had an aim of killing two million feral cats across Australia, but without any commitment to monitoring the impacts and outcomes, was criticized for not being based on sound scientific principles [176], while the second strategy did not have any target [178]. Feral cat control must be planned according to scientific principles and understanding of cat and prey biology, demography, and population dynamics [179,180]. Management programs that focus only on the control of feral cats without addressing habitat changes, such as loss of ground cover, are likely to be ineffective [174]. There are currently few feasible ways of reducing feral cats at large spatial scales, even on large islands [55]. The lethal control of feral cats is likely to be counter-productive because cats become wary, money can be expended on irrelevant targets, and efforts wasteful unless done in a synergistic way, such as in fenced enclosures with fire management and other species management [176,179]. Hunting cats to protect threatened species, such as by Indigenous hunters, also needs to be examined further for effectiveness [181]. The monitoring of the outcomes of feral cat control is essential, including the benefits to the native species affected by feral cats [162,176,179,182].

There are many other feral species that require attention, but here we address only one more, briefly. Feral pig management has been undertaken on Aak Puul Ngantam (APN) lands on Cape York by Wik people for a number of years (Justin Perry, pers. comm.). The early findings have revealed that, although feral pigs do a lot of damage to the country, their impacts vary according to the ecosystem, vegetation type, soil type, soil moisture content, season, and species. For instance, most fauna appear to be only lightly affected by the presence and activities of feral pigs, but both marine and freshwater turtles can be severely affected by feral pigs by predation [183]. The targeted management of pigs to prevent damage is vital, particularly where terrestrial turtles bury themselves during the

dry season and when both marine and terrestrial turtles are nesting. Feral pig management also requires good planning and monitoring.

### 3.4. Diseases

Disease may be another cause of declines in mammalian fauna, but the prevalence or importance of disease is poorly known [29,184]. Previous research in Australia of the impacts of diseases on native mammals has been limited but has identified that diseases are likely to have influenced the declines of a number of mammals [185–196]. Feral cats, dogs, pigs, cattle, horses, and rats may carry diseases, such as toxoplasmosis, giardia, cryptosporidium, and others, which can affect native mammals [162,186,192,197–199] and lead to death. The evidence of the effects of diseases, such as toxoplasmosis having an effect on native mammals is, however, scarce and remains to be a hypothesis for many species [200]. Epidemics and deaths, caused by disease, are notoriously difficult to detect when the evidence, in the form of carcasses, disappear from view very quickly after death [201–203]. Bats, such as flying foxes, are known vectors of disease [204–208] but the role of other native mammal species is less studied. More research is required in order to examine the disease prevalence in different native species in northern Australia, and which could be triggered to become epidemics in stressed populations of mammals.

### 3.5. Climate Change and Habitat Loss

Synergies among extinction drivers are important, as are synergies between species (co-extinctions) [20,29,209] and the threats that are identified here are likely to be exacerbated by the rapidly warming climate. The tropics are particularly vulnerable to the effects of climate change, simply because their climates are already warm and are the first to move out of the present climate range. The tropics are predicted to experience more extremes under the current warming trend, with resultant dangerous consequences for the tropical biota [59,210]. Northern Australia is experiencing higher intensity rainfall events, higher intensity cyclones, greater variability in rainfall, and higher temperatures above those already experienced, and the trend will continue [211]. Temperatures in places, such as Broome and Darwin, are likely to exceed 35 °C for two-thirds of the year [212,213]. The analysis of climate change in the Gulf of Carpentaria region has demonstrated that long-term temperature rises since 1910, when records began, has been in the order of 1.5 °C [214]. Predictions of major changes in climate, including temperature extremes, changed rainfall, and cyclonic events, have been made for other regions as well [212,214–220]. For example, the Bureau of Meteorology [130] has reported the highest number of days (44 in 2020, 45 in 2019) above 35 °C, along with other record-breaking temperature extremes for Darwin.

Global warming presents an immediate threat to species [211,221] and the tropical species may be affected more than other species due to their narrower heat and humidity tolerances and living closer to their thermal limits [222,223]. It has been suggested also that invasive species may be advantaged by climate change [224]. Research into the effects of regional warming on species and ecosystems in north Australia is poor, even for endangered species, such as the spectacled flying-fox [52], and often has not been identified as a priority driver of declines, even in recent syntheses [51]. Modelling on Wet Tropics species has shown that species declines are likely with changed climates [225], and declines associated with climate change are already being observed [226,227].

Beyond the physiological impacts on fauna, climate change is also likely to result in habitat change, and for some species habitat degradation and a reduction in available resources. A recent example of climate change impacting ecosystems in northern Australia has been the loss of mangroves in the Gulf of Carpentaria in 2016 along 1000 km of coast from the effects of climate change—lowered sea levels, lowered rainfall, and higher temperatures [228]. The mangrove loss was coincident with a massive coral bleaching event on the Great Barrier Reef [228]. The effects of climate change on mangrove habitat are possibly severe, but the consequences will not be known for years.

One of the main drivers of faunal declines is habitat loss. Modelling of extinction risk demonstrates that fragmentation of habitat, in contrast to a monotonic reduction in range, is more likely to plunge species in each remnant fragment below the minimum population viability levels [209]. Although clearing in northern Australia is not yet as extensive as in the more southerly regions of the continent, recent clearing of large areas of forest and woodland, such as in the Daly River area of the NT [229,230], in parts of north Queensland, such as on the Gilbert River (~50,000 ha on one property [231]), and other pending applications for clearing, are of concern.

## 4. Filling the Monitoring Gaps

### 4.1. Broadening the Research and Monitoring Network

Monitoring biodiversity and ecosystems is essential to support rigorous, evidence-based policies and decision making around environmental use and management, in addition to the following: measure environmental performance; trigger management actions to protect and maintain biodiversity and ecosystems; assess whether management actions work; and communicate with the public about ecosystems and biodiversity and their management [12,232–236] and is, therefore, core to adaptive management [237–239]. There is a need for greater accountability for species declines and failure to recover species, and to guard against complacency [240]. The investments in management practices need monitoring in order to ensure that they are not wasted [241]. The lack of monitoring is exacerbated by the lack of long-term research into population dynamics of north Australian mammals [61].

The main problems with understanding the status of north Australian fauna are as follows:

- Very few systematic studies of fauna exist that establish baselines and monitor trends in abundance and distribution over time;
- Most studies are short-term, poorly designed, or incompatible, survey limited areas of habitat (poor coverage), are poorly coordinated, do not make data available adequately, do not report adequately, do not link well with management, and do not adequately examine demographics [242];
- Studies have concentrated in only a few areas and on a few species, leaving very large areas and most species unstudied.

The long-term (more than 24 years) monitoring in the 'three-parks' studies that has informed much of the understanding of the declines was designed in order to monitor the suite of species present at the sites, which is necessary in order to understand faunal assemblages and population trends [22,23,29,243]. Standard monitoring surveys, however, have been found to be relatively poor in detecting trends that are useful to managers [13]. Existing monitoring of threatened species has focused on just a few [242,244], the majority in the Northern Territory and Western Australia. Recent efforts to understand the trends in threatened species have utilized expert elicitation in order to understand status and predict the declines of vertebrate species [47], but this is no substitute for on-ground data and more comprehensive monitoring is needed to inform management practices in an effort to prevent further degradation and decline [245–247]. Prioritizing research and monitoring investment is essential in a resource-constrained environment where government department funding has been cut dramatically [17] and government-sponsored research and monitoring programs have reduced substantially across much of Australia, e.g., [248].

Only three of the bioregions have adequate (or any) long-term monitoring of biodiversity, all in the NT and all in National Parks or Indigenous lands (Darwin Coastal, Pine Creek, Arnhem Plateau) [249] (Figure 3). Queensland savannas have no long-term biodiversity monitoring sites. A set of sites in the Desert Uplands bioregion in central Queensland was established in 2004 and research was conducted until about 2012, but the Long-Term Ecological Research Network (https://www.ltern.org.au/ (accessed on

29 April 2021)) that supported the studies was defunded in 2018, after less than six years of operation.

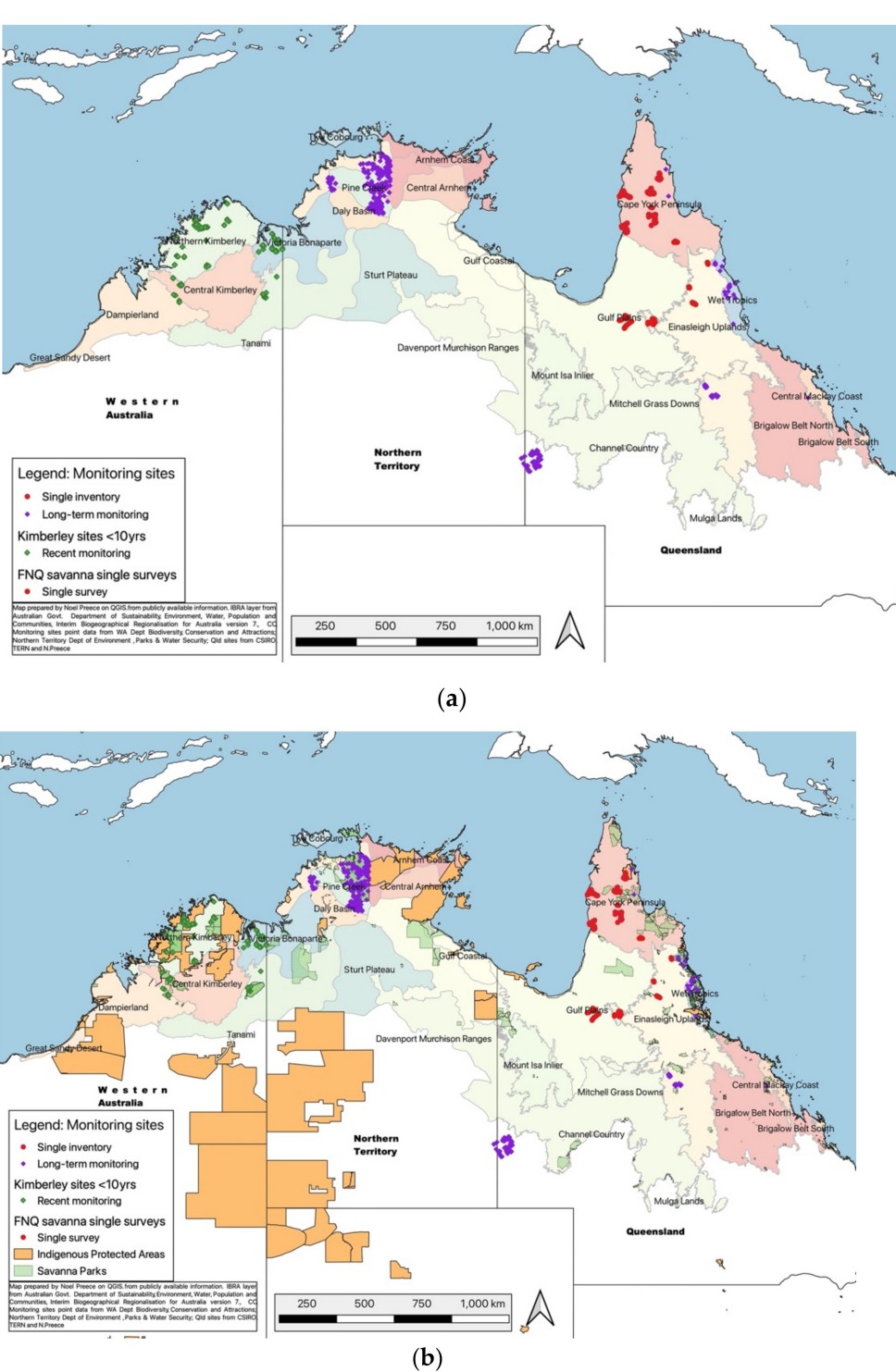

(**a**)

(**b**)

**Figure 3.** Monitoring sites across the north Australia savanna region, showing (**a**) IBRA bioregions, and (**b**) public protected areas ('Savanna Parks') and Indigenous Protected Areas. Sites marked with purple diamonds are long-term monitoring sites, green diamonds are medium-term monitoring sites, and red circles are single survey sites (sites do not include private conservation areas). The Wet Tropics (coastal north-east Queensland) are excluded as there are monitoring and research sites in the region operated by a number of institutions.

As northern Australia is extensive and diverse, and species ranges and genetic diversity are variable, determining trends for all of the species across their ranges should not depend on these few sites. Monitoring and research sites should be established across major savanna bioregions of north Australia in order to provide informed decisions on species recovery.

### 4.2. Adjusting Monitoring Methods

Decades of research and monitoring of fauna, particularly mammals, using what are called standard or conventional trapping and survey methods [39,236,250–255] have informed our understanding of faunal status and trends. But these methods are limited by scientific expertise, the ability of surveys to detect occupancy, the presence and trapability of mammals, frequency and density of trapping surveys, and the statistical power of survey programs. There is a limited number of scientists researching biodiversity in northern Australia, due mostly to inadequate resources, leaving little capacity to undertake further essential work. A pragmatic approach is therefore required.

Our first recommendation is that the revised methods that have been developed from more than 24 years of monitoring that have informed faunal declines across the northern Australian savannas (i.e., [31]) should be adopted across the region. The revised methods have taken into account the power of the statistical results so that they are more reliable. The revised methods also recognize that the previous methods sample some taxa poorly, including rare and threatened species, cryptic species, and some taxa, such as amphibians.

The revisions to methods recognize that it is essential that power analyses be conducted [256]. Recent analyses of the NT long-term studies of fauna and flora show that the statistical power has been inadequate [13,249]. Findings from this study were as follows:

- High confidence in the results of monitoring (statistical power of 0.8) could be achieved for moderate to large declines in only the most common and easily detected species;
- It was relatively poor for species with moderate occupancy and detectability, unless simulated declines were very large;
- For species with very low occupancy and detectability, no monitoring was able to detect even severe declines [13].

One of the few published studies to document mammal species on Cape York recognized these problems in north Queensland [39]. In order to improve the power of the monitoring studies, sites need to be monitored more frequently—for example every 3–5 years—and placing sites in areas of high occupancy and detectability [13].

Conducting monitoring using the revised standard methods is valuable for obtaining pilot data for power analysis [13] and to provide baseline inventories of species distributions and abundances and the opportunity to confidently identify species that might otherwise not be identified correctly. Handling animals caught in traps is also engaging and helps people to become more familiar with species, some of which are very difficult to see without trapping. Camera trapping provides identification certainty for only some of the larger species, but provided that these limitations are stated explicitly, can provide valuable information (e.g., [76]) (see Supplementary File S1). Einoder and colleagues provided an indication of the likely costs per annum for 50 sites (~500,000 AUD/a, plus establishment costs of ~300,000 AUD for survey equipment; [31]). Due to the absence of monitoring sites in north Queensland, this region would be a priority for establishing monitoring sites on both national parks and Indigenous lands, and other lands, such as private conservation lands and pastoral stations where possible.

Our second recommendation is to combine western scientific methods and Indigenous ecological knowledge [66,257–261] where appropriate. Such approaches have the potential to overcome the paucity of biodiversity monitoring sites across the savanna region while meeting Indigenous aspirations [262]. Much of the intact and threatened biodiversity occurs on Aboriginal lands, such as Indigenous Protected Areas (IPAs), which comprise more than 46% of Australia's National Reserve System and are often the only lands, apart from national parks [263] and privately protected areas [264], impacted minimally by

agriculture and grazing and therefore holding the remnant populations of threatened species. Indigenous ranger programs are already established with dedicated workforces and some operational funding, and experience with biodiversity surveys, such as on Warddeken, Wunambal-Gaambera, and Olkola lands. Many Indigenous rangers have worked extensively with western scientists for decades, and because many of the IPAs have active ranger groups working on country, they could present an invaluable resource for monitoring and there is potential for a mutual benefit for science and local communities from potential and actual collaborative surveys and subsequent analyses [265]. Many retain substantial knowledge of and a vital interest in their country and the plants and animals that they live alongside, and the activities undertaken by Indigenous people are likely 'to have benefits for threatened species on the basis . . . of a cultural connection to an area' [266]. The engagement of ranger groups makes sense, with the caveat that to engage appropriately means that a fair and equitable relationship must be developed with each ranger group on terms that do not impose unrealistic burdens and expectations on the rangers and traditional custodians of country [113,261,262].

The monitoring projects conducted by Indigenous rangers and others requires a revision of methods and depend on Indigenous support and greater levels of involvement [236,267]. The programs need to be relevant to the social and environmental values of Indigenous people, and conventional methods need to be adapted so that they are understandable and accessible; culturally acceptable; reflect the biodiversity and cultural values that are important to the Indigenous managers; are technically feasible for them to undertake; and are highly participatory [236].

Third, during the course of the research and consultation for this project, both Indigenous and non-Indigenous people suggested a focus on culturally important species [114], rather than the broader range of often cryptic species that western scientists find interesting and important in research, which will still be trapped e.g., [268]. Indigenous rangers have been utilizing, and have more potential to utilize, their skills to monitor culturally important taxa [114,269], such as kangaroos, wallabies, bettongs, koalas, food and culturally important plants, and other taxa. On Fish River Station (lands of the Labarganyin, Wagiman, Kamu, and Malak Malak people), northern quolls, black-footed tree-rats, northern brown bandicoots, common brushtail possums, dingoes, antilopine, common wallaroos, agile wallabies, emus, Australian bustards, and partridge pigeons were all identified as having cultural and conservation importance [236].

This approach can help to re-engage younger rangers and elders who may hold traditional knowledge of these species. Approaches similar to this have been reiterated in the Healthy Country Plans prepared for Indigenous lands across the region [118–122]. Working with culturally significant species can bring with it cultural and totemic constraints about sharing and publishing aspects of the findings about those species during research (as suggested by some Indigenous people consulted for this project) but negotiations about how the research is conducted, and how the results of research into species and ecosystems are reported and published, can resolve most issues before the research is undertaken [261,270].

It is up to the Indigenous custodians of the cultural knowledge of species whether or not to publish the culturally significant components [271], but it is important that the knowledge of species and their ecology, status, trends, and plight should be published because in Australia's democratic and economic system, evidence is one of the major driving forces for funding to manage and protect those species. Species' management and recovery plans depend on peer-reviewed scientific information [272,273] and publishing information about species raises the profiles of those species and improves consultation and cooperation with other workers in the field.

Finally, the monitoring sites must be selected according to multiple criteria and designed for long-term monitoring, using the best sites on offer in the regions proposed for monitoring. Negotiations with landholders and agreements with parks and wildlife

authorities for the use of sites and firm commitments and agreements need to be in place, and this will take time and resources [31].

*4.3. Scientific Permit Requirements for Live Trapping or Interference with Fauna*

A significant factor in fauna survey and monitoring is the requirement of each state and territory for 'wildlife' and 'scientific' permits and approvals from ethics committees to do the surveys. These requirements are established in legislation and policies as 'permits to take, kill, interfere with or use' fauna and other natural resources (the terms vary slightly among each state and territory). These rules are established for the welfare of the animals, and they apply to camera traps where baits are used to attract animals to the cameras. The inherent customary rights of Indigenous people on their traditional lands to hunt and fish and gather traditional foods and fibers do not obviate them from needing to obtain permits to trap and survey fauna for non-traditional purposes. Permittees usually, but not exclusively, have to hold a scientific degree in the relevant fields, such as zoology, biology, or ecology, and obtain references from recognized experts in the field of fauna studies. These can be onerous requirements. In all cases that we are aware of in northern Australian Indigenous lands and ranger groups, the permits are obtained and held by 'coordinators' of ranger groups or consultants, researchers, or government scientists, the vast majority of whom are non-Indigenous. This circumstance creates a dependency on the permit holder that some Indigenous rangers and groups may not find acceptable and needs to be further investigated.

## 5. Monitoring and Regional Employment

There are a number of essential requirements to fill the gaps in monitoring that are presented above, and it is a complex process that requires thought, planning, and commitment, including at the political level [274]. Identifying capacity alone is not sufficient—the people who have the potential must be engaged and committed, with finances and resources to undertake the work, continuity to enable job security and futures, and support of the wider scientific community in order to enable adequate research and monitoring [274].

We need to establish comprehensive reporting and data acquisition, uploading and curation systems [19], particularly focused on Indigenous observers' needs. We need to design the best monitoring approaches, aligned perhaps with the modified approaches developed for the NT sites after 20+ years of research [31,275] that reduce the number of sites per study region, but increase the intensity and frequency of monitoring. To begin with, monitoring sites could be established in the bioregions where there is a robust history of field management of fire, pests, and rehabilitation, and extended later as resources and facility increase. Many of the savanna burning programs have been established for a number of years and could be extended in order to enable biodiversity monitoring, given sufficient resources. A support network of scientists needs to be engaged to provide the essential scientific underpinnings of this work. State and Territory governments and universities could provide this network, which would need to be coordinated at the national level so that there is consistency of methods across borders. At the same time, we need to recognize and address the changes wrought by the application of the western ways of doing things, such as higher education through tertiary institutions that are located a long way from customary lands and people, and the Indigenous ways of learning and teaching, to try to avoid or ameliorate the problems of divided power-relationships in Indigenous communities.

Finally, adequate financial and associated resources are required in order to implement the monitoring programs. This will require at least doubling the budget back to 2013 levels, around several hundred million AUD per year in order to implement the program in north Australia, based on the assessments of needs for funding threatened species recovery across Australia [16]. Targeted investment to establish and test some of the proposed monitoring in areas of established conservation work would guide future investment.

While this review is focused on northern Australia, we consider that the approach and lessons could apply to many countries where mammalian diversity and its threats are underestimated due to the lack of available funds, irregularity of monitoring programs and the lack of statistical robustness, and the reduced engagement of citizen science.

**Supplementary Materials:** The following supporting information can be downloaded at: https://www.mdpi.com/article/10.3390/d14030158/s1, Supplementary file S1: Potential survey methods for Indigenous lands and other areas. References [276–295] are cited in the supplementary materials.

**Author Contributions:** Conceptualization, N.P. and J.F.; methodology, N.P.; formal analysis, N.P.; writing—review and editing, N.P. and J.F. All authors have read and agreed to the published version of the manuscript.

**Funding:** Funding for the original study was provided by The Nature Conservancy.

**Institutional Review Board Statement:** Not applicable.

**Acknowledgments:** This report relied on consultations with a number of people who are leading the research in northern Australia. Not all experts were available for consultation at the time. Thanks to David Hinchley, Luke Preece, Brett Murphy, Graeme Gillespie, Ian Radford, Alexander Watson, Ben Corey, Luke Einoder, Brydie Hill, Teresa Eyre, Stephen van Leeuwin, Sharon Prior, Mike Heywood, Jeremy Russell-Smith, Terry Mahney, Debbie Symonds, Tom Vigilante, Justin Perry, Alys Stevens, and Emilie Ens. Thanks to Penny van Oosterzee for constructive comments on the draft. We acknowledge the Traditional Custodians of the many nations and lands on which we work, and pay our respects to the Elders past and present.

**Conflicts of Interest:** The authors declare no conflict of interest.

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
