# Peer review of "Gaps in Monitoring Leave Northern Australian Mammal Fauna with Uncertain Futures"

_diversity, doi:10.3390/d14030158_

Round 1

Reviewer 1 Report

The authors presented a manuscript aimed at locating and assessing gaps in monitoring schemes for animal populations in northern Australia. Their second aim was to propose a new scheme that would adequately resolve problems and estimate population trends.

This is an important task, as effective monitoring is critical for the conservation of animal populations. The authors have exhaustively presented available knowledge on the subject. However, the chosen style and analyses are very long, overly descriptive and prosaic (too much detail), thus failing to convey the study aims to the reader. Nor the current gaps in monitoring schemes were precisely obvious and understood, neither the future proposed schemes were effectively explained and presented.

Comments

Manuscript title (lines 2-3)

The authors should be specific about which faunal species they referred to. It seems that mammal species, and not all of them were the target of the manuscript.

Abstract (lines21-27)

These are long sentence, hard for the reader to follow. Please split to shorter sentences. Also, the whole manuscript contains long sentences, even longer than these two. See for example lines 40-44, 48-53, 55-64, 101-108. Please check and revise throughout.

In-text references (lines 37-38 and throughout)

Please follow Journal style for in-text references.

Study aims (lines78-94)

(1) Study aims seem a little too vague and intertwined. Please revise to strictly follow what was studied. One would expect to see, considering the manuscript title: (1) an identification and assessment of the gaps in monitoring and (2) a proposed framework for addressing these gaps. See also later comments.

(2) It is stated that an emphasis on mammal was given. This is different to the reference to the fauna in the title. Please be specific. Chose a term, e.g., small mammals, and use it throughout for consistency.

(3) The only reference to methodology is found here. The authors must be more explicit. Give more information in a Methodology section. For example, about databases searched, keywords used, number of other information sources, number of articles located, number of articles deemed useful by category and used in the analyses.

Section 2 (lines 95-290)

(1) This is the monitoring literature review section. It is a large essay, containing so much information that ends up in tiring and confusing the reader. The text should be largely eliminated. The authors mix, in many details, population trends, their reliability and possible reasons for this reliability, and also discuss gaps in monitoring. Relevant information should be presented in a Table. This table should contain: the monitored species and entries for each species concerning study area (rather broadly to avoid very many entries, e.g., western Australia), monitoring method, population size or/and trend, reliability of the estimate, adequacy of the monitoring scheme, and the sources of the information. Information contained in the table should be summarized in the text, i.e., number of species by study area, number of species with reliable estimates that increased, were stable or declined. In doing so, the reader would readily understand the situation (gaps in monitoring, a main aim, and also population trends and their reliability) and would be able to follow next sections. In this section, text highlighting important issues, such as species conservation status, taxonomic issues and comments on important gaps in monitoring should be also given (e.g., frequency, methods, false trends, failure to locate rare species), in a succinct way.

(2) Box 1 (line 118) also contains very detailed text. It has no scope as a box, relevant information should be given, largely condensed, in the text.

Section 3 (lines 291-655)

A very large essay about the causes of population declines. It seems largely irrelevant here. The reader would be confused and driven away from the scope of the manuscript. After understanding the gaps in monitoring schemes, one would expect to see a proposed framework to address them. Eliminate this text. Authors could include a paragraph or so in the previous section, about the hows, whys and consequences of the causes of population declines. Also, Figure 1 is not about quolls.

Sections 4 and 5 (lines 656-893)

(1) Again here, the authors chose a descriptive style to explain their proposed strategy towards better monitoring schemes. Although clearer than the previous sections, further improvement would allow for a more thorough understanding. A main omission is the lack of a diagram. The authors should present their proposed framework diagrammatically, in a flow chart. Inputs and desired outputs should be identified and connected. What are the proposed phases (e.g., gap identification, strategic planning, stakeholders, methods, timeframe)? What are the groups that would participate to monitoring? What are the organizations that would support (with funds, legislation, permits, other) monitoring? What is the necessary time framework? What are the proposed methods (if any, otherwise propose their suitable identification)? What are the desirable outcomes (reliable estimates, informed and efficient conservation)? Who would benefit? How are these connected or affected by each other? These are some of considerations to be included in the proposed framework for effective monitoring. In the text, each of the framework’s components should be briefly discussed and ways for their implementation identified and proposed.

(2) It is not clear who produced Figure 3. Also, Fig. 3 a has no legend.

(3) In several parts of the manuscript the authors refer, as responsible for the monitoring to indigenous people, to rangers, to responsible groups. It is not clear, especially to foreign readers, if indigenous people are also the rangers, or some of them. Please clarify this when first arise in the text.

Author Response

Reviewer comment: The authors presented a manuscript aimed at locating and assessing gaps in monitoring schemes for animal populations in northern Australia. Their second aim was to propose a new scheme that would adequately resolve problems and estimate population trends. This is an important task, as effective monitoring is critical for the conservation of animal populations. The authors have exhaustively presented available knowledge on the subject. However, the chosen style and analyses are very long, overly descriptive and prosaic (too much detail), thus failing to convey the study aims to the reader. Nor the current gaps in monitoring schemes were precisely obvious and understood, neither the future proposed schemes were effectively explained and presented.

 Author response:

 We thank the reviewer for their comments on this article. We acknowledge that the article is exhaustive and felt this was necessary as it benchmarks an important stage in research and monitoring for the region as a whole. A previous article (Ziembicki et al 2015) was similar in scope with much detail and has proved to be a valuable reference point in the current circumstance of increasing declines of fauna in the region. Most of the detail we feel is necessary for scholars and policy-makers so that they are well-informed about the current situation.

On the second point about the current gaps in monitoring schemes not being precisely obvious, we have mapped and referenced the actual monitoring locations in the text and on the figures. To go into more detail for a region where monitoring is and would be undertaken by multiple institutions would be presumptuous on our part. Identifying the very large gaps in monitoring across the region provides scholars and policy-makers with the peer-reviewed basis for designing a sound monitoring scheme or program. Indeed, part of the problem for the region is cross-jurisdiction boundary disagreements and disputes about approaches, methods, resourcing and the like (a dispute which we have avoided in this article).

Reviewer comment: Manuscript title (lines 2-3) - The authors should be specific about which faunal species they referred to. It seems that mammal species, and not all of them were the target of the manuscript.

 Author response:

We have changed the title to state mammals as these are the species most at risk. The fauna survey methods however trap and identify all vertebrate taxa within study sites and these are equally monitored and recorded with mammals.

Reviewer comment: Abstract (lines21-27) - These are long sentence, hard for the reader to follow. Please split to shorter sentences. Also, the whole manuscript contains long sentences, even longer than these two. See for example lines 40-44, 48-53, 55-64, 101-108. Please check and revise throughout.

 Author response:

We have shortened a number of sentences throughout the text in accordance with the reviewer’s suggestions. We have left the abstract as is, because we could not see how to break the sentences in a logical way, and felt that the sentences were not overly long in the abstract anyway. We have addressed the length of sentences where there were two ideas in the sentence, but the others were long because the second part of the sentence was related to the first part as a clarification, or contrary view and the sentence as constructed made logical sense. The journal’s endnote referencing style has also now helped improve the readability of sentences.

Reviewer comment: In-text references (lines 37-38 and throughout) - Please follow Journal style for in-text references.

Author response: Thanks, yes we have now revised to journal style. This should also in part address readability of some sentences and perceived length of certain sentences and sections.

Reviewer comment: (1) Study aims seem a little too vague and intertwined. Please revise to strictly follow what was studied. One would expect to see, considering the manuscript title: (1) an identification and assessment of the gaps in monitoring and (2) a proposed framework for addressing these gaps. See also later comments.

Author response:

We have been specific in what we set out to do: (1) update and present the range of issues around faunal declines in the region to provide context to the gaps in monitoring; (2) articulate the gaps in knowledge to better frame future monitoring of species and ecosystems; and (3) suggest an approach to improve monitoring and research, identifying some of the priority areas and the scale of the resources needed. We are therefore unclear what the reviewer meant here.

Reviewer comment: (2) It is stated that an emphasis on mammal was given. This is different to the reference to the fauna in the title. Please be specific. Chose a term, e.g., small mammals, and use it throughout for consistency.

Author response:

We have been more specific about the target taxa: ‘an emphasis on small to medium size (under 35-5,500 g) mammals’. Small-medium mammals are the taxa at most risk of decline and from the existing studies are showing the most rapid and alarming declines. We had included all taxa in the general discussion as when trapping and surveying for fauna we survey all vertebrate taxa and report on them. Trends in other taxa have been mentioned in this article, but to expand more on other taxa would create a much longer article.

Reviewer comment: (3) The only reference to methodology is found here. The authors must be more explicit. Give more information in a Methodology section. For example, about databases searched, keywords used, number of other information sources, number of articles located, number of articles deemed useful by category and used in the analyses.

Author response:

We have expanded on our methodology in the text as follows:  ‘This article reviews and synthesizes research on and monitoring of threatened vertebrate species of northern Australia, with an emphasis on small to medium size (under 35-5,500 g) mammals. We first reviewed the literature on the biodiversity of northern Australian savannas, with an emphasis on that occurring between 2010 and 2021 and interviewed many of the ecologists working in the savannas to identify additional published articles, grey literature, and active monitoring projects and programs. We each maintain comprehensive bibliographies of research on fauna in northern Australia from our combined six decades of working in the region and so, with the additional benefit of our close links with research colleagues in the region, a strict primary ‘literature review’ was deemed unnecessary.’

The review we have conducted was for the last decade and based strongly on information gathering by conducting interviews with many of the leading researchers working in northern Australia to ensure that we had all the current information and publications on faunal declines for the region. We have worked in northern Australia a combined six decades or so, and have researched and published with many of the other researchers doing work in the region throughout that period. There are only a few dozen researchers working on this subject, and we would argue that we know them all or nearly all. We each also maintain a comprehensive bibliography and collection of papers, and link with most of the journals with key words to alert us to new articles published on the broad field of fauna research. Undertaking a primary literature review of the nature suggested would be a trivial and superfluous exercise that is likely to produce no new work other than what we already have. While the reviewer’s suggestion might be valid for fields less familiar to us (and we have undertaken similar primary reviews for other topics), our expertise and long-experience and direct connections make a primary review far less relevant.

Reviewer comment: Section 2 (lines 95-290) (1) This is the monitoring literature review section. It is a large essay, containing so much information that ends up in tiring and confusing the reader. The text should be largely eliminated. The authors mix, in many details, population trends, their reliability and possible reasons for this reliability, and also discuss gaps in monitoring. Relevant information should be presented in a Table. This table should contain: the monitored species and entries for each species concerning study area (rather broadly to avoid very many entries, e.g., western Australia), monitoring method, population size or/and trend, reliability of the estimate, adequacy of the monitoring scheme, and the sources of the information. Information contained in the table should be summarized in the text, i.e., number of species by study area, number of species with reliable estimates that increased, were stable or declined. In doing so, the reader would readily understand the situation (gaps in monitoring, a main aim, and also population trends and their reliability) and would be able to follow next sections. In this section, text highlighting important issues, such as species conservation status, taxonomic issues and comments on important gaps in monitoring should be also given (e.g., frequency, methods, false trends, failure to locate rare species), in a succinct way.

Author response:

We agree with reviewer 1 that this is a large essay, but differ with their assessment and proposals. We have added a table on the status of mammals across northern Australia that updates and expands on table 1 published by Ziembicki et al 2015 in order to address one of the issues raised. We have examined the text related to this section and believe that none is superfluous as it provides a summary with references of the main research and monitoring developments that have been published since the reference papers. This detail is important for fellow researchers to know what has been published and what changes have occurred, as noted by Reviewer 2.

The purpose of the article, however, is not to provide a synthesis of the species monitored and their status. Many species are not monitored and the point of the article is to demonstrate what is known about species for which there is published information, which includes their status and trends and then to highlight the substantial gaps in monitoring. To do what is being suggested in this comment would produce a very different paper, which would summarise all details on all species monitored, and the adequacy of that monitoring, and to summarise all the other species for which there is no monitoring, which is a much longer list, and defeats the purpose of bringing the information on known species declines up to date and therefore to inform decision-makers and researchers on the significant gaps in monitoring and knowledge. We have been careful to not attempt a synthesis of all the species’ status and monitoring gaps. Indeed, part of the purpose of the paper is to demonstrate that there is little known about many species outside the two main areas with long-term monitoring (i.e. Top End of the Northern Territory and the Kimberley regions). On one of the points raised by reviewer 1, there is no information on population size for nearly every species in northern Australia. For most species, only trends are modelled. Only when populations have crashed are there estimates of numbers for a few species, such as the Spectacled Flying-fox, which we discuss in the article.

Reviewer comment: (2) Box 1 (line 118) also contains very detailed text. It has no scope as a box, relevant information should be given, largely condensed, in the text.

Author response:

 Box 1 consists of three short paragraphs of summaries of work that has been done in the Queensland region, but with some unpublished data. It serves to illustrate the major gaps in monitoring across Queensland and would not flow well in the main text. We think it works better as a box, and would prefer to leave it as such.

Reviewer comment: Section 3 (lines 291-655) A very large essay about the causes of population declines. It seems largely irrelevant here. The reader would be confused and driven away from the scope of the manuscript. After understanding the gaps in monitoring schemes, one would expect to see a proposed framework to address them. Eliminate this text. Authors could include a paragraph or so in the previous section, about the hows, whys and consequences of the causes of population declines. Also, Figure 1 is not about quolls.

Author response:

 As explained in the introduction, this essay is intended to update three seminal papers, one by Ziembicki et al 2015, one by Woinarski et al 2011 and the other by Fitzsimons et al 2010, but not to replace them. In doing so, we intended to update some of the issues raised in those papers, including the causes and putative causes of declines. Since the publication of the earlier papers, there has been some significant research undertaken and some important findings made about causes that differ from those in the previously published articles, so we feel that it is essential that we provide a reasonably comprehensive review of this research. It is also important to put monitoring in context, as one of the main purposes of monitoring is to inform management and responses.

We corrected figure 2’s caption.

Reviewer comment: Sections 4 and 5 (lines 656-893) (1) Again here, the authors chose a descriptive style to explain their proposed strategy towards better monitoring schemes. Although clearer than the previous sections, further improvement would allow for a more thorough understanding. A main omission is the lack of a diagram. The authors should present their proposed framework diagrammatically, in a flow chart. Inputs and desired outputs should be identified and connected. What are the proposed phases (e.g., gap identification, strategic planning, stakeholders, methods, timeframe)? What are the groups that would participate to monitoring? What are the organizations that would support (with funds, legislation, permits, other) monitoring? What is the necessary time framework? What are the proposed methods (if any, otherwise propose their suitable identification)? What are the desirable outcomes (reliable estimates, informed and efficient conservation)? Who would benefit? How are these connected or affected by each other? These are some of considerations to be included in the proposed framework for effective monitoring. In the text, each of the framework’s components should be briefly discussed and ways for their implementation identified and proposed.

Author response:

We agree with the reviewer’s suggestion for further improvement, but feel this is beyond the scope of two researchers to achieve in this article. Doing what is suggested would take at least a workshop of people committed to monitoring, and our experience is that this would be some considerable task. There are literally hundreds of stakeholders among many institutions and organisations and across jurisdictional boundaries. Previous attempts to standardize survey and monitoring methods have failed to adequately resolve the differences, in our personal experience of vegetation mapping and fauna survey methods. The three states, for instance, use ‘standard’ fauna survey methods that are not necessarily consistent nor compatible, although consistency has increased in recent years.

The purpose of the paper is to review the most recent trends in mammal fauna to update the key papers mentioned, and to identify and highlight the major gaps in monitoring. This task alone has not been done before at the level we have attempted. Lack of monitoring has been identified by others, but not at the regional and more specific level.

Reviewer comment: (2) It is not clear who produced Figure 3. Also, Fig. 3 a has no legend.

Author response: The authors produced Figure 3. We have now added the legend to both maps

Reviewer comment: (3) In several parts of the manuscript the authors refer, as responsible for the monitoring to indigenous people, to rangers, to responsible groups. It is not clear, especially to foreign readers, if indigenous people are also the rangers, or some of them. Please clarify this when first arise in the text.

Author response:

We have made a note on this in the text on page 5.

Reviewer 2 Report

Review of Diversity-1583140

The present review is well-written and presented and provides evidence that regular, long-term and well-funded monitoring programs integrating local people and the scientific community are necessary for deciphering the threats, pressures and extinction tendencies of mammals, in Australia and elsewhere.

The authors expose their arguments correctly and support their findings with an extensive literature, mainly focused on Australia. Nevertheless, it would be useful to provide data on the percentages of threatened mammals, and especially those that apparently face imminent extinction, per taxonomic group. In this way it will be visible which groups are more threatened and the number of taxa involved. This will strengthen their arguments for funding for long-term monitoring and subsequent management implementation where necessary.

Concerning the northern quoll, the authors tackle a sensitive issue of problematic taxonomy ta the subspecies level that is necessary for unravelling populations that need to be conserved and those that are apparently healthy. This is even more important when translocation projects are scheduled. It would be interesting if the authors could provide some more thoughts in this issue and what could be a good practice (chapter 2.4).

The effect of global warming on the increase of species extinction is very complex. The authors have mainly tackled the direct effects on physiology and tolerance of species but there are indirect ones related to habitat change, degradation, reduction of available resources, etc. It would be nice if the authors would comment on these issues too.

Culturally important species may be a good way to involve local communities and engage new recruits in conservation activities, but most of them represent common species that were easily spotted and trapped. This leaves other, equally important, more elusive species in their shade. On the other hand, if trends of extinction are traced in the former, this is representative of a general faunal decline of a certain area and may thus serve as flag species that will attract policy makers.

What is exposed in this review applies to most countries in the world where mammalian biodiversity and its threats are underestimated due to lack of available funds, irregularity of monitoring programs and lack of statistical robusticity, and the reduced engagement of citizen science. All these things are discussed in this manuscript, which, albeit focused on the problems of Northern Australia, can have a global relevance.

Some minor points:

Legend of figure 2 is the same as (and corresponds to) figure 1. Please amend.

Line 474 Australia

Line 487 delete the second “and”

Author Response

Reviewer comment: The present review is well-written and presented and provides evidence that regular, long-term and well-funded monitoring programs integrating local people and the scientific community are necessary for deciphering the threats, pressures and extinction tendencies of mammals, in Australia and elsewhere.

Author response:

Thanks you for this supportive comment and your constructive criticisms.

Reviewer comment: The authors expose their arguments correctly and support their findings with an extensive literature, mainly focused on Australia. Nevertheless, it would be useful to provide data on the percentages of threatened mammals, and especially those that apparently face imminent extinction, per taxonomic group. In this way it will be visible which groups are more threatened and the number of taxa involved. This will strengthen their arguments for funding for long-term monitoring and subsequent management implementation where necessary.

Author response:

We have provided a table that updates Ziembicki et al 2015 with data from Woinarski et al 2018 on all the threatened mammal species known from northern Australia. The table shows mammal taxa by family, the status of each, and an assessment of the status of monitoring for each. We think this will satisfy the suggestion made.

Reviewer comment: Concerning the northern quoll, the authors tackle a sensitive issue of problematic taxonomy at the subspecies level that is necessary for unravelling populations that need to be conserved and those that are apparently healthy. This is even more important when translocation projects are scheduled. It would be interesting if the authors could provide some more thoughts in this issue and what could be a good practice (chapter 2.4).

Author response:

In the section on Northern Quolls, we cited IUCN/SSC 2013 which addresses this issue. We have added the words “and they should comply with the IUCN/SSI guidelines”. We feel that providing much more than this would require a much longer argument because of the complexities and nuances inherent in threatened species recovery and translocations. We also note Reviewer 1 considered this section should be shortened.

Reviewer comment: The effect of global warming on the increase of species extinction is very complex. The authors have mainly tackled the direct effects on physiology and tolerance of species but there are indirect ones related to habitat change, degradation, reduction of available resources, etc. It would be nice if the authors would comment on these issues too.

Author response:

We have now acknowledged this in a new sentence in the manuscript. However, as there are many articles on the effects of climate change on habitats, degradation, available resources, and we feel that we had to focus on some of the key elements related to the mammal species we are considering.

The new sentence/s read:

“Beyond the physiological impacts on fauna, climate change is also likely to result in habitat change, and for some species habitat degradation and a reduction of available resources. A recent example of climate change impacting ecosystems in northern Australia has been the loss of mangroves in the Gulf of Carpentaria in 2016 along 1000 km of coast from the effects of climate change – lowered sea levels, lowered rainfall and higher temperatures (Duke et al. 2017).”

Reviewer comment: Culturally important species may be a good way to involve local communities and engage new recruits in conservation activities, but most of them represent common species that were easily spotted and trapped. This leaves other, equally important, more elusive species in their shade. On the other hand, if trends of extinction are traced in the former, this is representative of a general faunal decline of a certain area and may thus serve as flag species that will attract policy makers.

Author response:

Thanks again for this comment. We have re-organized the section to prioritize standard and targeted faunal monitoring, and then to address culturally important species. We agree that it is critical that monitoring addresses the suite of species in any area and that culturally important species comprise an important added dimension to the more ‘traditional’ focus species. The manuscript didn’t originally make that clear and we trust our revision now does.

Reviewer comment: What is exposed in this review applies to most countries in the world where mammalian biodiversity and its threats are underestimated due to lack of available funds, irregularity of monitoring programs and lack of statistical robusticity, and the reduced engagement of citizen science. All these things are discussed in this manuscript, which, albeit focused on the problems of Northern Australia, can have a global relevance.

Author response:

We agree. Is the reviewer suggesting that we add words to this effect in the ultimate paragraph, as we added one using words similar to reviewer’s suggestions? We can find no review article on the inadequacy of monitoring globally, so maybe this could present an opportunity. We have added the following sentence here:

“While this review is focussed on northern Australia, we consider that the approach and lessons could appliy to many countries where mammalian diversity and its threats are underestimated due to lack of available funds, irregularity of monitoring programs and lack of statistical robustness, and the reduced engagement of citizen science.”

Reviewer comment: Legend of figure 2 is the same as (and corresponds to) figure 1. Please amend.

Author response: Now corrected, thank you

Reviewer comment: Line 474 Australia

Author response: Now corrected, thank you

Reviewer comment: Line 487 delete the second “and”

Author response: Deleted, thank you

Round 2

Reviewer 1 Report

The authors have successfully addressed all suggestions. The manuscript can now be proposed for publication.